# Metabolomics analysis of human acute graft-versus-host disease reveals changes in host and microbiota-derived metabolites

David Michonneau [1,2,9], Eleonora Latis[3,9], Emmanuel Curis [4,5,6], Laetitia Dubouchet [2], Sivapriya Ramamoorthy[7], Brian Ingram[7], Régis Peffault de Latour[1,2], Marie Robin[1], Flore Sicre de Fontbrune[1], Sylvie Chevret[6,8], Lars Rogge [3,10] & Gérard Socié [1,2,10]*

Despite improvement in clinical management, allogeneic hematopoietic stem cell transplantation (HSCT) is still hampered by high morbidity and mortality rates, mainly due to graft versus host disease (GvHD). Recently, it has been demonstrated that the allogeneic immune response might be influenced by external factors such as tissues microenvironment or host microbiota. Here we used high throughput metabolomics to analyze two cohorts of genotypically HLA-identical related recipient and donor pairs. Metabolomic profiles markedly differ between recipients and donors. At the onset of acute GvHD, in addition to host-derived metabolites, we identify significant variation in microbiota-derived metabolites, especially in aryl hydrocarbon receptor (AhR) ligands, bile acids and plasmalogens. Altogether, our findings support that the allogeneic immune response during acute GvHD might be influenced by bile acids and by the decreased production of AhR ligands by microbiota that could limit indoleamine 2,3-dioxygenase induction and influence allogeneic T cell reactivity.

[1] Hematology Transplantation, Saint Louis Hospital, 1 avenue Claude Vellefaux, 75010 Paris, France. [2] Université de Paris, INSERM U976, 75010 Paris, France. [3] Institut Pasteur, Immunoregulation Unit, Department of Immunology, 25 rue du Docteur Roux, 75015 Paris, France. [4] Université de Paris, INSERM UMR-S1144, 75013 Paris, France. [5] Université de Paris, Laboratoire de biomathématiques plateau iB2EA 7537–BioSTM, Faculté de pharmacie, 75006 Paris, France. [6] Service de Biostatistique et Information Médicale, Hôpital Saint-Louis, AP-HP, 1 avenue Claude Vellefaux, 75010 Paris, France. [7] Metabolon, Inc., Morrisville, NC, USA. [8] Université de Paris, INSERM U1153, Epidemiology and Biostatistics Sorbonne Paris Cité Research Center (CRESS), ECSTRA Team, 75010 Paris, France. [9] These authors contributed equally: David Michonneau, Eleonora Latis. [10] These authors jointly supervised this work: Lars Rogge, Gérard Socié. *email: gerard.socie@aphp.fr

Allogeneic hematopoietic stem cell transplantation (HSCT) is a major treatment for hematologic malignancies and for inherited or acquired hematopoiesis disorders. However, it is still hampered by high morbidity and mortality rates[1] mainly due to graft-versus-host disease (GvHD). Although insight to GvHD pathophysiology has been gained from the development of animal models[2], most of these experimental studies have focused on the role of different T cell subsets[3,4]. However, it has been recently demonstrated that the allogeneic immune response might be influenced by external factors such as tissues microenvironment[5,6] or host microbiota[7,8]. Recently, metabolomics has emerged as a major new field in system biology that can reflect how genetics, environmental factors, or microbiota affect host biochemical processes[9]. During the transplantation process, patients are subjected to chemotherapy, antibiotics, and antiviral therapy that can alter tissue biology and microbiota, and could impact GvHD severity or relapse risk[10,11]. Many metabolites can contribute to immune response regulation through their direct influence on immune cell activation, proliferation, or survival[12–14]. Hence, it has been suggested that pre-transplant metabolomics profiles could impact transplant outcomes[15]. Microbiota-derived metabolites such as short-chain fatty acids could regulate tissue reparation and immune cell activation[16–18]. In mice, butyrate-producing bacteria can mitigate experimental acute GvHD[16] and a higher abundance of these bacteria in humans is associated with a reduced rate of respiratory viral infection after allogeneic-HSCT[19]. If many metabolites could regulate immunity, there is no clear broad and unbiased overview of how metabolomics pathways are influenced by the transplantation process and which changes are more specifically associated with acute GvHD at disease onset[20].

Using high throughput metabolomics to analyze two cohorts of genotypically HLA-identical-related recipients and donors, we show that HSCT is followed by major changes in metabolomics profiles of recipients. At acute GVHD onset, significant variation of host- and microbiota-derived metabolites are identified, mainly affecting indole compounds of the tryptophan metabolism, a group of metabolites that acts as ligands for the aryl hydrocarbon receptor (AhR), together with bile acids (BAs) and plasmalogens. This suggests that dysbiosis, together with transplantation-related alteration of host metabolism, induce major change in circulating metabolites in recipients that might influence allogeneic immune cell reactivity.

## Results

### Metabolomics profiling of patients after allogeneic-HSCT.
Herein we studied two cohorts of patients, one single center cohort from Saint Louis hospital (cohort 1, $n = 43$) and one multicentric from 13 French transplantation centers (cohort 2, $n = 56$), who received an allogeneic-HSCT from an HLA-identical sibling donor. Plasma samples from sibling donors were obtained before stem cell collection and thus considered as a healthy group of reference. Recipients' samples were collected at the onset of acute GvHD before any treatment with corticosteroids or at day 90 after transplantation for those who did not develop GvHD (Fig. 1a). Patient and transplant characteristics are described in Supplementary Data 1. Importantly, almost all patients had natural feeding at the time of sampling, excluding the possibility that metabolomics changes might be attributed to artificial parenteral or enteral nutrition. Using ultrahigh performance liquid chromatography-tandem mass spectrometry (UPLC-MS/MS) we detected 801 and 927 circulating metabolites in cohorts 1 and 2, respectively, with 653 metabolites being shared by both cohorts (Fig. 1b). The average distribution of these shared metabolites into each metabolic pathway was: lipid

(46.18%), amino acid (24.16%), xenobiotics (13%), nucleotides (4.43%), carbohydrate (3.82%), cofactors and vitamins (3.82%), peptide (3.21%) and energy (1.38%) pathways (Fig. 1c).

### Alteration of recipients' metabolome after transplantation.
To identify whether metabolome of allogeneic-HSCT recipients differs from those of healthy subjects, recipients without GvHD were compared to their related sibling donors. For each metabolite, UPLC-MS/MS generates many empty values when the corresponding sample's value is under the lower limit of quantification[21]. To identify metabolites that were more frequently detected in recipients or donors, irrespective of the amounts of compounds, the distribution of non-detectable values was assessed (Fig. 2a, Supplementary Fig. 1a). Using this approach, we identified five and three metabolites in cohorts 1 and 2 that were more frequently detected in donors or in their related recipients after correction for multiple testing, respectively (Supplementary Data 2 and 3). After filtering metabolites with more than 50% of non-detectable values and imputation with half of the minimal detected values (Fig. 2b, Supplementary Fig. 1b), metabolites with putative biological relevance were identified by comparison of the amount of each metabolite between recipients without GvHD and their paired related donors (Fig. 2c, d, Supplementary Fig. 1c, d, Supplementary Data 4 and 5). As compared with healthy subjects, allogeneic-HSCT recipients without GvHD were mainly characterized by a significant increase in the amounts of complex lipid metabolites, especially fatty acid, mono and diacylglycerol and primary BA. Amino acid metabolism was also altered by transplantation. Especially, tryptophan-derived metabolites and taurine production were strongly decreased after transplantation, whereas polyamine metabolites were increased in recipients without GvHD. Global analysis of metabolites behavior by comparison of their relative amount confirmed that BA and indolepropionate were the most affected after transplantation (Fig. 2e, f, Supplementary Fig. 1e, f). Among the polyamine group, we observed a significantly increased level of 5-methylthioadenosine (MTA) and N-acetylputrescine that might play a role in protection of gut integrity[22], could inhibit macrophage activation[23], and reduce T cell activation[24,25]. Interestingly, it was also recently demonstrated that intestinal inflammation is regulated by microbial-derived metabolites through the regulation of NLRP6 inflammasome signaling. Whereas taurine may activate inflammasome signaling and lead to production of the pro-inflammatory cytokine, polyamine may inhibit this pathway and protect gut from auto-inflammation[26]. Bile acids have recently been described as a potent regulator of the immune system through the inhibition of the NLRP3-dependent inflammasome pathway[27], the recruitment of NKT cells in the liver, through the regulation of CXCL16 production by liver sinusoid endothelial cells[28,29], and to reduce macrophages activation, migration, and cytokines production[30–32]. Altogether, these results suggest that the metabolomics changes observed in patients without GvHD could contribute to decrease inflammasome activation and could thus protect epithelial integrity after transplantation. Using the same approach, we compared metabolomic profiling of recipients with GvHD to their related donors. Few metabolites were more frequently detected in donors or in recipients with GvHD ($n = 0$ in cohort 1 and $n = 6$ in cohort 2) (Fig. 3a, Supplementary Fig. 2a, Supplementary data 6). Comparison of the amounts of each metabolite identified 150 and 182 metabolites that were significantly changed after transplantation, in cohorts 1 and 2 respectively, with 110 metabolites shared by both cohorts (Fig. 3c, d, Supplementary Fig. 2c, d, Supplementary Data 7 and 8). Most of metabolic pathways involved in recipients with GvHD were similar to those identified

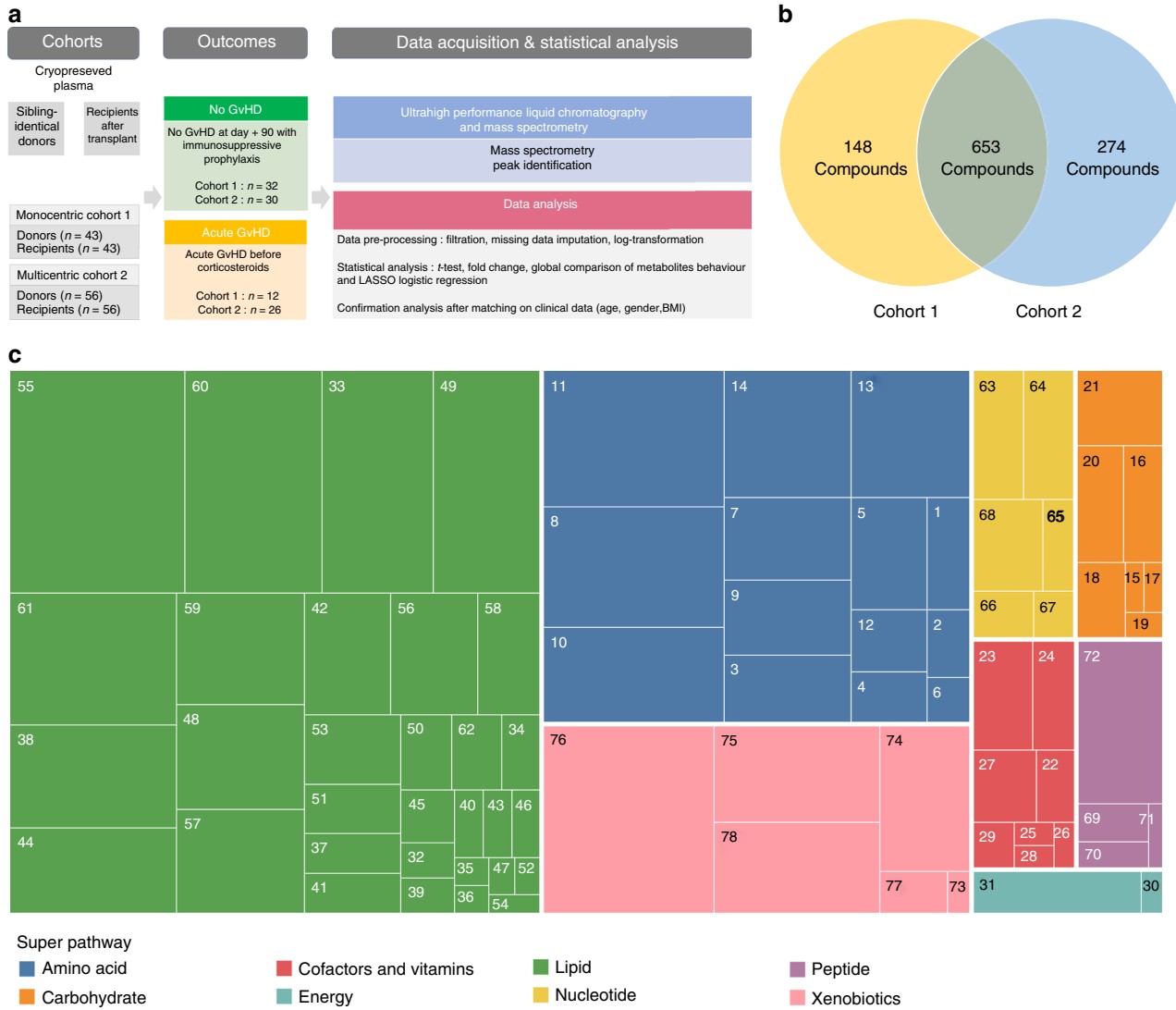

**Fig. 1 Study design and general overview of detected metabolites. a** Metabolomic profile was studied in two cohorts of patients, a single center cohort from Saint Louis hospital, Paris (cohort 1), and a multicentric French cohort (cohort 2). Plasma samples were collected in sibling donors before stem cell collection and in recipients after transplantation, at day 90 for those who did not developed GvHD or before corticosteroid treatment at disease onset for those who developed GvHD. Samples were frozen and then analyzed by ultrahigh-performance liquid chromatography-tandem mass spectrometry (UPLC-MS/MS) for circulating metabolites identification and quantification. *One donor did not give his consent for further analysis. **b** In all, 801 and 923 metabolites were identified in cohorts 1 and 2, respectively, with 653 metabolites shared by both cohorts. **c** Treemap showing the distribution of metabolites among super pathway (in color) and their related sub-pathways (square) for the 653 metabolites used for statistical analysis. Each number corresponds to one of the sub-pathway listed in Source Data file.

in recipients without GvHD, with the exception of primary BA and mono/diacylglycerol that were increased in recipients by comparison with their donors. A common feature of all recipients was a strong decrease in xenobiotics detection, especially in xanthine, tobacco, and food metabolites. This suggests that all recipients modified their behaviors regarding food or tobacco intake, as strongly recommended after HSC transplantation.

**Acute GvHD is characterized by specific metabolomics changes.** As recently demonstrated in mice models, metabolites could regulate GvHD severity[17,33]. In patients, metabolome mainly reflects the metabolism of host cells and of its microbiota. To determine which metabolomics changes are associated with acute GvHD, we compared the metabolome of recipients with GvHD at disease onset with that of recipients without GvHD. Assessment of non-detectable value distribution among patients revealed that indolepropionate, a microbial-derived compound from tryptophan

metabolism, was the only metabolite significantly less frequently detected at GvHD onset after Bonferroni correction in cohort 2 (Fig. 4a, Supplementary Fig. 3a, Supplementary Data 9 and 10). Comparison of metabolites quantities revealed substantial changes in amino acids (tryptophan and arginine pathways) and lipids (lysoplasmalogens, plasmalogens, and phospholipids) that were all decreased in patient with GvHD. In addition to the above described pathways, we also identified a strong increase of multiple complex lipid products in patients with acute GvHD, such as medium- and long-chain fatty acid, polyunsaturated fatty acid (n3 and n6) and in primary and secondary BA (Fig. 4b, c, Supplementary Fig. 3b, c, Supplementary Data 11 and 12). Global comparison of metabolites ratio between patients also confirmed that BA, plasmalogens, tryptophan, and arginine metabolites were the main contributors of the changes observed at GvHD onset (Fig. 4e, f, Supplementary Fig. 3e, f). These results were confirmed after adjustment for age, gender, and BMI that were considered as the main confounding

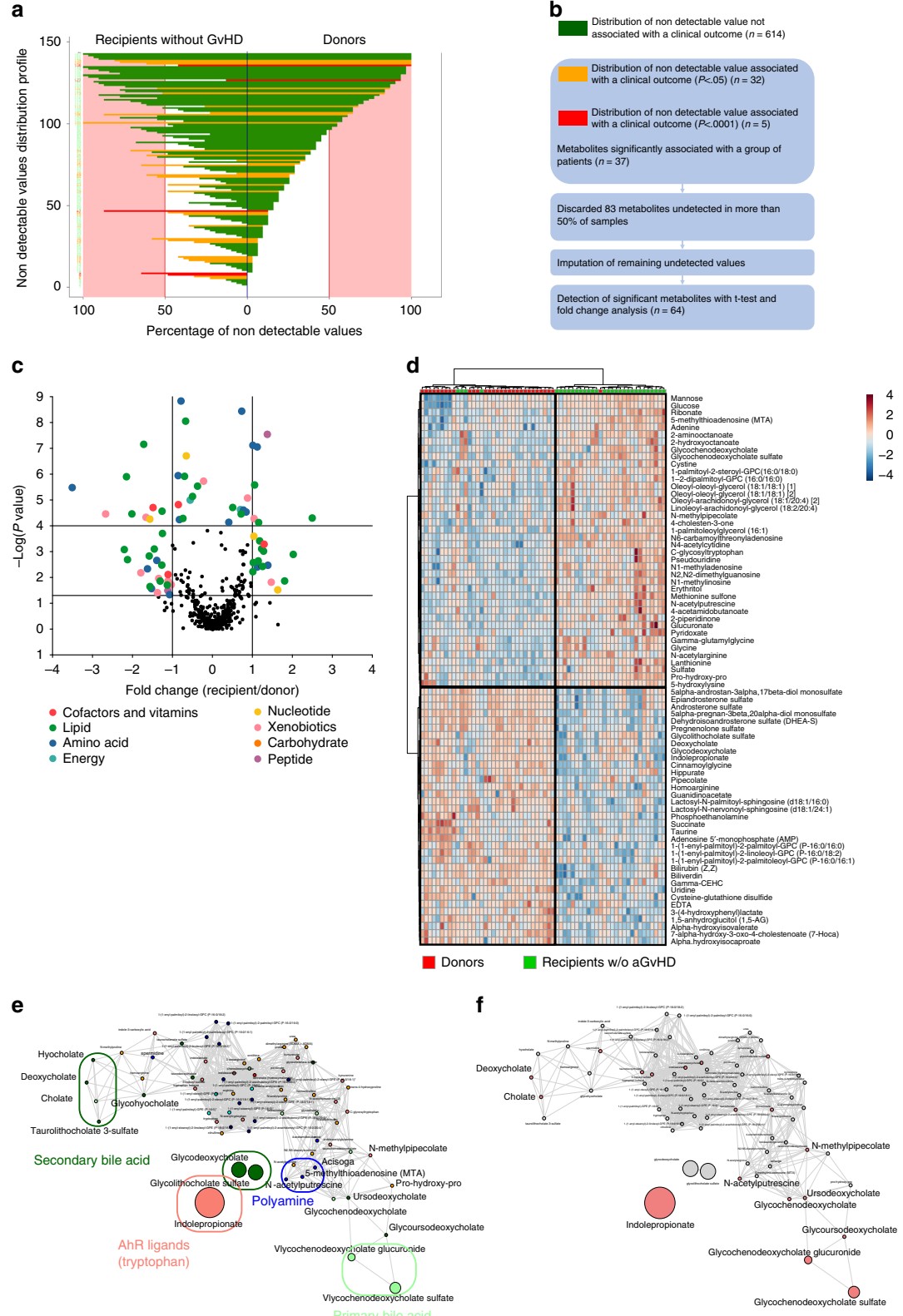

factors. Thus, we were able to identify a group of metabolites that were still significantly associated with GvHD onset in both cohorts (Supplementary Data 13, Fig. 5). To confirm the implication of these metabolites at acute GvHD onset, PCA was used to confirm that recipient with or without GvHD could be discriminated in multivariate analysis (Fig. 6a, b). Metabolites that mostly contribute to acute GvHD profile were then identified with sparse PLS-DA

(Fig. 6c, d, Supplementary Data 14) and used to build an over-representation analysis (ORA) of the main pathways that contribute to GvHD (Fig. 6e, f). This approach confirmed that most of metabolites identified in univariate analysis were also selected by sPLS-DA and belong to the previously identified metabolic pathways, especially plasmalogens, arginine, and tryptophan metabolism. Considering the fact that metabolites variation might result

**Fig. 2 Metabolomics changes after transplantation in recipients without GvHD. a** Distribution of non-detectable values for each metabolite was assessed in recipient without GvHD and in their related paired donors. Each line represents a profile of distribution for non-detectable value between groups, observed for at least one metabolite. The frequency of non-detectable values in both groups was compared for each distribution profile using a McNemar exact test and colored as follows: green: frequency of non-detectable values non-significantly different between groups; orange: frequency of non-detectable values significantly different between groups with $p < 0.05$ but not significant after Bonferroni correction for multiple testing; red: frequency of non-detectable values significantly different between groups after Bonferroni correction for multiple testing ($p < 7.69 \times 10^{-5}$). Red area represents the threshold used for filtration of metabolites with more than 50% of non-detectable value. **b** A total number of 651 metabolites were analyzed for comparison of paired donor and recipients without GvHD. Among them, 37 metabolites were more frequently detected in one group of patients, including 5 metabolites that were still significant after Bonferroni correction (Supplementary Data 2). Eighty-three metabolites with more than 50% of non-detectable values were discarded for further analysis, and remaining non-detectable values were replaced after imputation with half of the minimal detected value for each corresponding metabolite. **c** The remaining 568 metabolites were compared with a paired Student test followed by a Bonferroni correction. The volcano plot represents the variation of metabolites amount between recipients without GvHD and their related paired donors according to the $-\log(p$ value). List of the 64 significant metabolites is available in Supplementary Data 4. **d** Heatmap representation of the most significant metabolites after Student test, after hierarchical clustering of samples. **e, f** Significant variation of metabolites was confirmed after global comparison of the relative amounts between compounds. Main pathways identified in the previous analysis (i.e. polyamine metabolism, tryptophan metabolism, urea cycle, arginine & proline metabolism, bacterial or fungal, plasmalogen or lysoplasmalogen, and primary or secondary bile acid metabolism) were used to build an undirected graph where each node is a metabolite and two nodes are connected if their ratio is unchanged between the two groups (see Methods). The same network was first colored according to the considered sub-pathway (**e**) and then according to microbial-derived metabolites (red nodes) (**f**).

from their interdependency, lasso logistic regression was performed to identify the set of metabolites that seems to be predictive of acute GvHD (Supplementary Data 14). Using lasso regression analysis, we were able to confirm that both tryptophan and arginine pathways were involved, especially the AhR ligand 3-indoxyl sulfate that was detected in both cohorts. We finally compared compounds rates for these pathways and observed similar profile in both cohorts at GvHD onset (Fig. 7a). In addition to host-derived metabolites, we also identified significant variation in microbiota-derived indole compounds in both cohorts, especially in AhR ligands (Fig. 7b). All these microbiota-derived metabolites changes are consistent with a major dysbiosis associated with acute GvHD[7,34,35]. Interestingly, it was recently demonstrated that fecal microbiota transplantation in HSCT patients was followed by increased microbiome diversity and a concomitant increase in the level of urinary 3-indoxyl sulfate[36].

## Discussion

GvHD is an immune reaction from donor immune cells targeting allogeneic antigens in recipients, whose pathophysiology is still only partially understood in humans. Recent experimental researches have highlighted the complex network of interactions and regulations between immune cells, microbiota, and host environment[4]. Here we report that patients who underwent allogeneic-HSCT have major metabolomics changes compared to healthy subjects, and that acute GvHD onset seems to be associated with specific differences in metabolomics profile by comparison with patients who did not developed GvHD. Comparing patients without GvHD with those who developed an unpredictable time onset GvHD leads to intrinsic differences between groups of patients that may have contributed to the observed variations in metabolites. These differences include slight differences in the day of sampling after transplantation, feeding, or treatments received at the time of sampling. To minimize the putative impact of these confounding factors, we used two independent cohorts of patients to confirm our results and explored groups of as much as possible similar patient characteristics in terms of HLA matching, immunosuppression, or diet mode. Our results suggest that transplantation is followed by modification in the recipients' eating behaviors, mainly affecting xanthine metabolism (suggesting different consumption of chocolate, tea, and coffee) or condiments-derived metabolites (piperine, alliin, cinnamoilglycine). Importantly, most patients had oral feeding at time of sampling and did not receive artificial feeding that could have affected metabolism.

At the onset of acute GvHD, we observed a significant decrease in tryptophan metabolites, including microbiota-produced compounds, such as 3-indoxyl sulfate, indoleacetate, indoleacetylglutamine, and indolepropionate, and host-derived compounds produced by the IDO (indoleamine 2,3-dioxygenase) tolerogenic pathway[37], N-acetylkynurenine and picolinic acid. Indole compounds are AhR ligands that regulate IDO induction in immune cells[38], whereas kynurenine was recently demonstrated to be a strong inhibitor of T cell activation[39]. However, it should be emphasized that the functional consequence of AhR ligands decrease should be explored, best in animal models. It has been demonstrated experimentally that the biological effect of some AhR ligands, such as indolepropionate, is not dose-responsive[40,41], suggesting that the absence of some AhR ligands could be more relevant than their decrease in acute GvHD. This is also why it appears that the comparison of metabolites that are detected or not detected in the recipients (Fig. 4a, Supplementary Fig. 3a, Tables 9 and 11) is, in our view, as important as the comparison of metabolites amount. Using this approach, we identify N-acetyl-kynurenine in both cohorts, as well as indole-3-carboxylic acid, 3-indoxyl sulfate, and indolepropionate in cohort 2, which were not only decreased in recipients with GvHD but also more frequently undetectable in these patients. Our findings thus support the hypothesis that acute GvHD onset is associated with a decreased production of AhR ligands by microbiota that could limit IDO induction. IDO activity has been associated with GvHD severity in mice and humans[42–44], and positively regulates AhR signaling through the production of the AhR ligand kynurenine. AhR can also modulate Th17 response and promote tolerance through the differentiation and the activation of Treg and Tr1 cells[45]. Otherwise, we observed significant variation in citrulline, a precursor of arginine synthetized by the intestine which has been considered as a surrogate biomarker of gut epithelial mass[46]. It was recently identified as a risk factor of acute GvHD, as a consequence of epithelial damage and gut permeability[47,48]. Many lipids pathways seem to be altered at the onset of acute GvHD. Polyunsaturated fatty acids were increased at the onset of GvHD and are the precursors of the eicosanoid family, such as leukotriene or prostaglandin (PG)[49]. Inhibition of 5-lipoxygenase (5-LO) reduces leukotriene B4 production from arachidonic acid and protects mice from acute GvHD in an experimental model[50]. 5-LO deficiency was associated with a decreased production of pro-inflammatory cytokines such as interferon-γ, TNF-α, and IL-17, and an increased level of circulating IL-10 (ref. [50]). In addition, PGE2 is critical for gut repair after inflammation and was recently found to be regulated by bacterial-

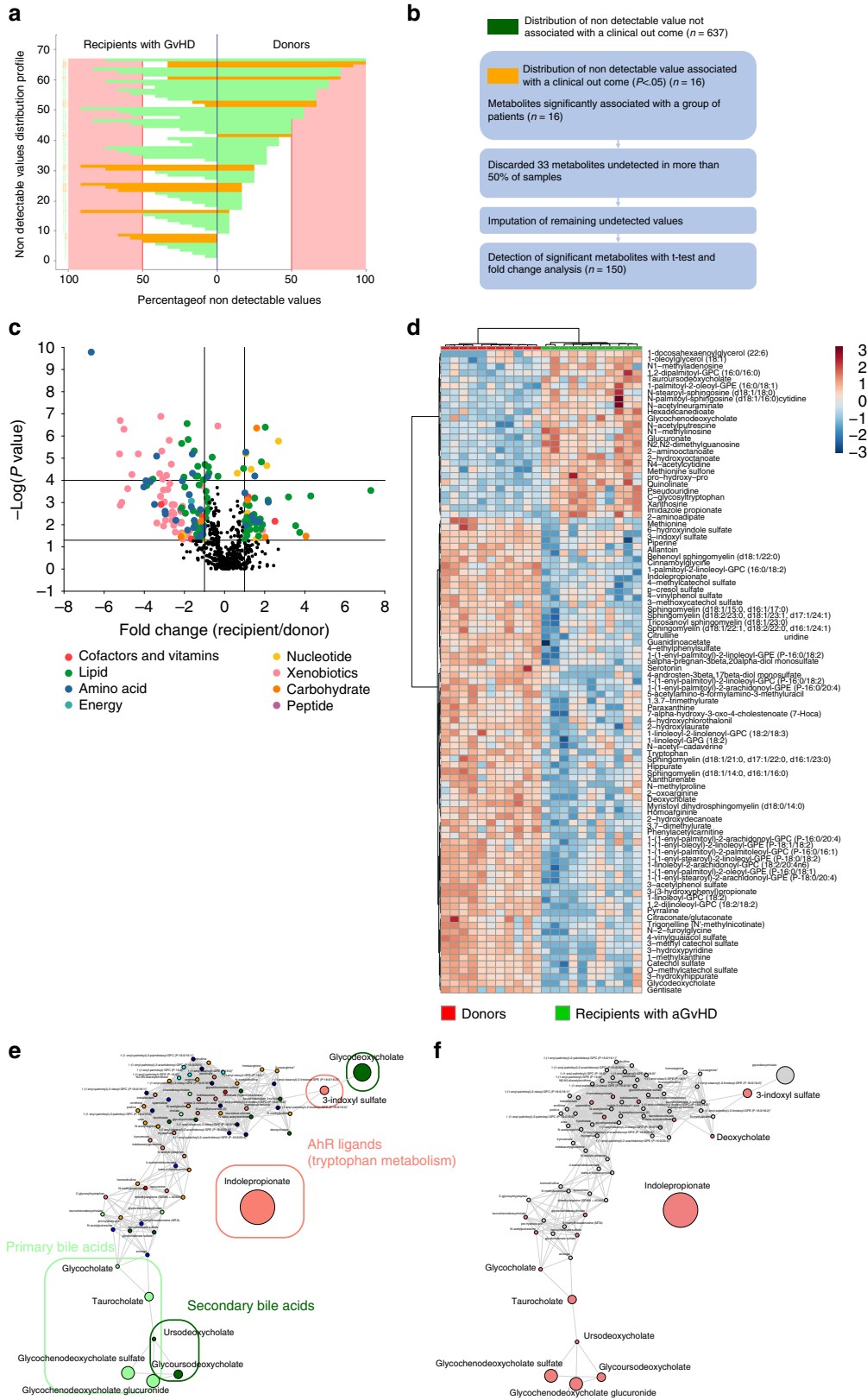

metabolized BA[51]. Bile acids effects on immune response and inflammation are not fully understood, with data suggesting a role in pro-inflammatory cytokines production, T cell activation and neutrophil recruitment[52,53], while other recent data suggest that they could inhibit inflammasome activation[27]. Finally, aGvHD was associated with a dramatically low level of plasmalogens and lyso-plasmalogens at disease onset. The biological role of plasmalogens in inflammatory processes remains controversial, with a putative protective effect against ROS products[54]. Interestingly, it has been suggested that plasmalogens could play a role as reservoir for

**Fig. 3 Metabolomics changes after transplantation in recipients with GvHD. a** Distribution of non-detectable values for each metabolite was assessed in recipient with GvHD and in their related paired donors. The frequency of non-detectable values in both groups was compared and colored as follows: green: frequency of non-detectable values non-significantly different between groups; orange: frequency of non-detectable values significantly different between groups with $p < 0.05$ but not significant after Bonferroni correction for multiple testing. Red area represents the threshold used for filtration of metabolites with more than 50% of non-detectable value. **b** A total number of 653 metabolites were analyzed for comparison of paired donor and recipients with GvHD. Among them, 16 metabolites were more frequently detected in one group of patients, including no metabolite that were still significant after Bonferroni correction. To compare the amount of each metabolite between recipients and their donors, 33 metabolites with more than 50% of non-detectable values were discarded for further analysis. **c** The remaining 620 metabolites were compared with a paired Student test followed by a Bonferroni correction. The volcano plot represents the variation of metabolites amount between recipients with GvHD and their related paired donors according to the $-\log(p\ \text{value})$. List of the 150 significant metabolites is available in Supplementary Data 7. **d** Heatmap representation of the most significant metabolites after Student test, after hierarchical clustering of samples. **e, f** Significant variation of metabolites was confirmed after global comparison of the relative amounts between compounds. Main pathways identified in the previous analysis (i.e. polyamine metabolism, tryptophan metabolism, urea cycle, arginine & proline metabolism, bacterial or fungal, plasmalogen or lysoplasmalogen, and primary or secondary bile acid metabolism) were used to build an undirected graph where each node is a metabolite and two nodes are connected if their ratio is unchanged between the two groups (see Methods). The same network was first colored according to the considered sub-pathway (**e**) and then according to microbial-derived metabolites (red nodes) (**f**).

second intracellular messengers such as arachidonic acid and subsequent production of eicosanoid, as it was recently demonstrated that plasmalogens hydrolysis in LPS-primed macrophages increases arachidonic release for eicosanoid synthesis[55].

An unsolved question raised by these results is the relation between acute GvHD and metabolome alterations at disease onset. It is tempting to consider that at least a part of metabolites changes could be involved in GvHD pathogenesis, as suggested by recent results in animal model[16,17]. Nevertheless, one cannot exclude that GvHD by itself could induce alteration of tissues metabolism. Indeed, it has been previously demonstrated, mostly in experimental models, that dysbiosis observed in GvHD could be both partly causative and consequence of the allogeneic immune response[7,16]. Our results suggest that many variations observed in metabolites could be due to microbiota changes after transplantation especially if GvHD occurs. Recently, it was demonstrated that plasma metabolome can predict gut microbiome α-diversity[56]. In this study, 40 metabolites were identified as being associated with human disease and microbiome, many of them being also identified after transplantation, our study. In our cohorts, all patients received antibiotics at the time of sampling. Antibiotics are involved in dysbiosis of microbiota and most likely dysbiosis participates to metabolomics changes after transplantation. However, whether this phenomenon could be a factor increasing acute GvHD severity remains to be explored. Interestingly, recent clinical studies suggested that antibiotics may have a major impact on GvHD-risk in humans[8,10].

Altogether, our results revealed that allogeneic-HSCT is associated with major metabolomics variations. The onset of acute GvHD was characterized by reduced production of tryptophan-derived metabolites, especially host- or microbial-derived AhR ligands that might contribute to increase allogeneic immune response in the recipient. Reduced production of plasmalogens together with the increased level of BAs and polyunsaturated acids are potential metabolomics pathways that could be involved in the early pro-inflammatory response during GvHD. These findings highlight major biological processes involved at GvHD onset that may represent new therapeutic targets for GvHD prophylaxis or treatment.

## Methods

**Patients.** Patients and their related donors analyzed in this study underwent an allogeneic HSCT in Saint Louis hospital, Paris, France (cohort 1, $n = 43$) or in one of the 33 French national transplant centers involved in CRYOSTEM Consortium, funded under the French Government's National Investment Program (Investissement d'avenir) (cohort 2, $n = 56$). Inclusion criteria were adult patients (more than 18-year-old), with an HLA-identical sibling donor who underwent an allogeneic-HSCT. Patients with HIV or HTLV co-infection were excluded. Our

objectives were to include at least 40 patients per cohort with an expected acute GvHD incidence of 40%. Clinical data were extracted from medical records and included gender, age, CMV status, underlying hematological diagnosis, HLA matching between donor and recipient, stem cell source, conditioning regimen, T cell depletion, GvHD prophylaxis, presence or absence of GVHD and nutrition type (oral vs. parenteral vs. enteral) at sampling day.

Donors' samples were collected during medical visit before any stem cell collection procedure. Recipients' samples were collected at day $90 \pm 5$ after transplantation for patients who did not develop acute GvHD at any time after transplantation, and at the onset of symptoms, before starting any corticosteroid treatment, for acute GvHD. Patients in the non-GVHD group never developed neither classical nor late onset acute GvHD. All samples were collected on EDTA tubes (BD Vacutainer, K3E 7.2 mg, Plus blood Collection Tubes), centrifuged within 4 h to collect plasma and immediately frozen at $-80\,°C$ until processing. In patient who had GvHD, organ involvement and stage, the date, the treatment, the response to treatment and the date of response were recorded. GvHD grade was evaluated according to Glucksberg score and Consensus criteria[57,58].

All patients gave their written consent for clinical research. This non-interventional research study with no additional clinical procedure was carried out in accordance with the Declaration of Helsinki. Data analyses were carried out using a database with all patient identifiers removed. This study was declared to the CNIL (Commission National Informatique et Liberté, number KoT1175225K) and was approved by the local ethic committee and Institutional Review Board (CPP Ile de France IV, IRB number 00003835).

**Samples collection and preparation.** Concerning CRYOSTEM plasma, samples annotated have been provided by the CRYOSTEM Consortium (https://doi.org/10.25718/cryostem-collection/2018) and the SFGM-TC (Société francophone de greffe de moelle et de thérapie cellulaire). Aliquots of 1 mL were divided in four aliquots of 250 μL for further study and sent to Metabolon company (Morrisville, US) for further process. Samples were prepared using the automated MicroLab STAR® system from Hamilton Company. Several recovery standards were added prior to the first step in the extraction process for QC purposes. For the metabolomic analysis, a total of 100 μL of sample was extracted under vigorous shaking for 2 min (Glen Mills GenoGrinder 2000) with methanol 80% containing the following recovery standards: DL-2-fluorophenylglycine, tridecanoic acid, d6-cholesterol, and DL-4-chlorophenylalanine. The resulting extract was divided into five fractions: two for analysis by two separate reverse phase (RP)/UPLC-MS/MS methods with positive ion mode electrospray ionization (ESI), one for analysis by RP/UPLC-MS/MS with negative ion mode ESI and one for analysis by HILIC/UPLC-MS/MS with negative ion mode ESI. The remaining aliquot was reserved for backup. Samples were placed briefly on a TurboVap® (Zymark) to remove the organic solvent. The sample extracts were stored overnight under nitrogen before preparation for analysis.

**Mass spectrometry.** All methods utilized a Waters ACQUITY UPLC and a Thermo Scientific Q-Exactive high-resolution/accurate mass spectrometer interfaced with a heated electrospray ionization (HESI-II) source and Orbitrap mass analyzer operated at $R = 35,000$ mass resolution. The sample extract was dried then reconstituted in solvents compatible to each of the four methods. For each sample, two aliquots of each sample were reconstituted in 50 μL of 6.5 mM ammonium bicarbonate in water (pH 8) for the negative ion analysis and another two aliquots of each were reconstituted using 50 μL 0.1% formic acid in water (pH ~3.5) for the positive ion method. Each reconstitution solvent contained a series of standards at fixed concentrations to ensure injection and chromatographic consistency. The internal standards consist of a variety of deuterium labeled or halogenated biochemicals specifically designed both to cover the entire chromatographic run and

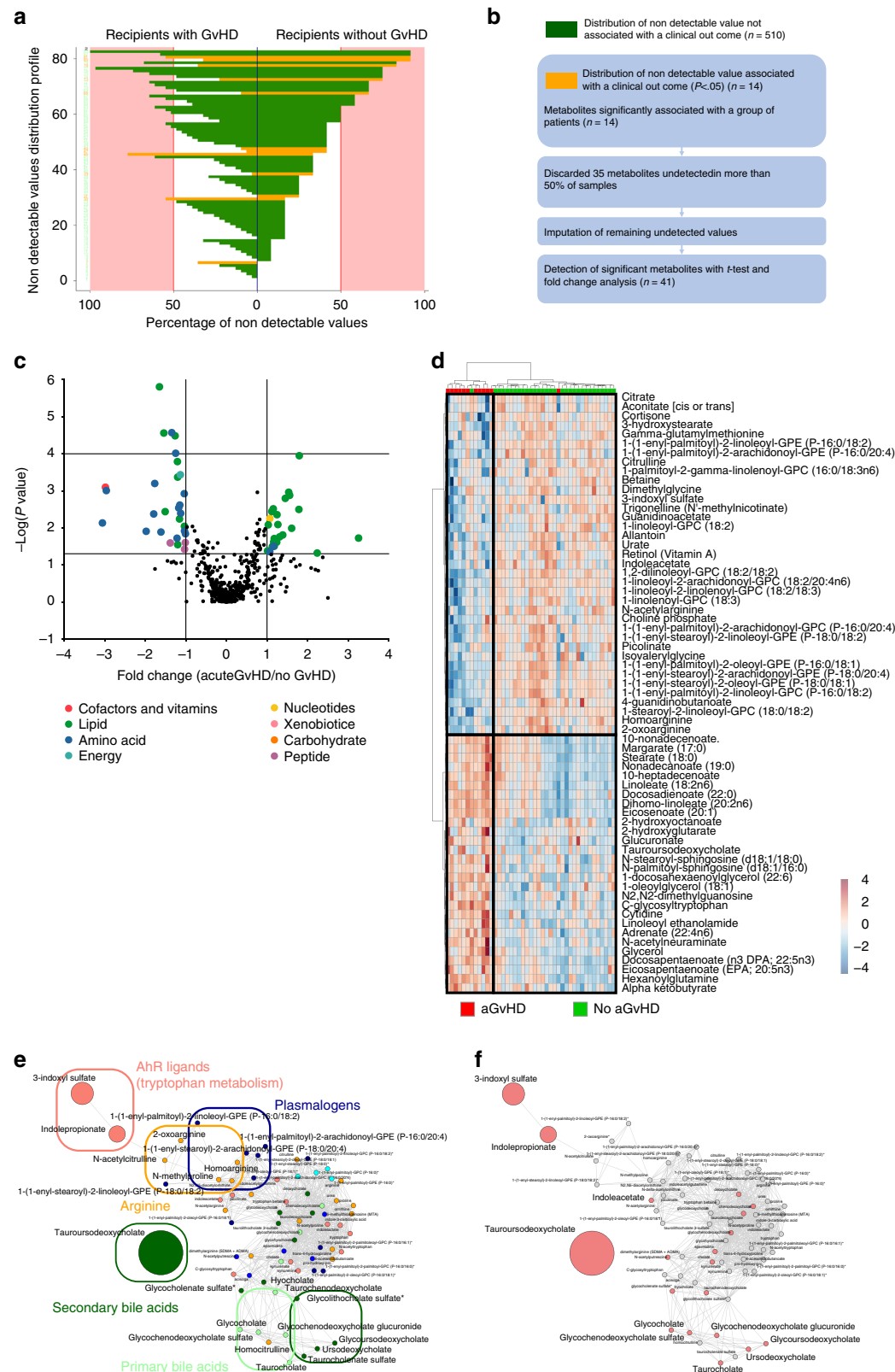

to not interfere with the detection of any endogenous biochemicals. Authentic standards of d7-glucose, d3-leucine, d8-phenylalanine, and d5-tryptophan were purchased from Cambridge Isotope Laboratories (Andover, MA). D5-hippuric acid, d5-indole acetic acid, and d9-progesterone were procured from C/D/N Isotopes, Inc. (Pointe-Claire, Quebec). Bromophenylalanine was provided by Sigma-Aldrich Co. LLC. (St. Louis, MO) and amitriptyline was from MP Biomedicals, LLC. (Aurora, OH). Recovery standards of DL-2-fluorophenylglycine and DL-4-chlorophenylalanine were from Aldrich Chemical Co. (Milwaukee, WI).

Tridecanoic acid was purchased from Sigma-Aldrich (St. Louis, MO) and d6-cholesterol was from Cambridge Isotope Laboratories (Andover, MA). Standards for the HILIC dilution series of alpha-ketoglutarate, ATP, malic acid, NADH, and oxaloacetic acid were purchased from Sigma-Aldrich Co. LLC. (St. Louis, MO) while succinic acid, pyruvic acid and $NAD^+$ were purchased from MP Biomedicals, LLC. (Santa Ana, CA).

Limit of detection (LOD) for standards analyzed in a dilution series using reverse phase chromatography is available in Supplementary Data 15.

**Fig. 4 Metabolomics pathways changes associated with aGvHD onset. a** Distribution of non-detectable values for each metabolite was assessed in recipients with or without GvHD. The frequency of non-detectable values in both groups was compared for each distribution profile using a Fisher exact test for a 2 × 2 contingency table and colored as follows: green: frequency of non-detectable values non-significantly different between groups; orange: frequency of non-detectable values significantly different between groups with $p < 0.05$ but not significant after Bonferroni correction for multiple testing. Red area represents the threshold used for filtration of metabolites with more than 50% of non-detectable value. **b** A total number of 524 metabolites were analyzed for comparison of recipients with or without GvHD. Among them, 14 metabolites were more frequently detected in one group of patients, but none was significant after Bonferroni correction (Supplementary Data 9). To compare the amount of each metabolite between recipients with or without GvHD, 35 metabolites with more than 50% of non-detectable values were discarded for further analysis. **c** The remaining 489 metabolites were compared with a Student test with Satterwhaite's correction for unequal variance, followed by a Bonferroni correction. The volcano plot represents the variation of metabolites amount between recipients with GvHD and those without GvHD according to the $-\log(p\ value)$. List of the 41 significant metabolites is available in Supplementary Data 11. **d** Heatmap representation of the more significant metabolites after Student test after hierarchical clustering of sample. **e, f** Significant variation of metabolites was confirmed after global comparison of the relative amounts between compounds. Main pathways identified in the previous analysis (i.e. polyamine metabolism, tryptophan metabolism, urea cycle, arginine & proline metabolism, bacterial or fungal, plasmalogen or lysoplasmalogen, and primary or secondary bile acid metabolism) were used to build an undirected graph where each node is a metabolite and two nodes are connected if their ratio is unchanged between the two groups (see Online Methods). The same network was first colored according to the considered sub-pathway (**e**) and then according to microbial-derived metabolites (red nodes) (**f**).

One aliquot was analyzed using acidic positive ion conditions (LC pos), chromatographically optimized for more hydrophilic compounds. In this method, the extract was gradient eluted from a C18 column (Waters UPLC BEH C18-2.1 × 100 mm, 1.7 μm) using water and methanol, containing 0.05% perfluoropentanoic acid (PFPA) and 0.1% formic acid (FA) at pH = 2.5. Elution was performed at 0.35 mL min$^{-1}$ in a linear gradient from 5% to 80% of methanol containing 0.1% FA and 0.05% PFPA over 3.35 min. A second aliquot was also analyzed using acidic positive ion conditions; however, it was chromatographically optimized for more hydrophobic compounds. In this method, the extract was gradient eluted from the same afore mentioned C18 column using methanol 50%, acetonitrile 50%, water, 0.05 % PFPA, and 0.01 % FA at pH = 2.5 and was operated at an overall higher organic content. Elution was performed at 0.60 mL/min in a linear gradient from 40% to 99.5% over 1 min, hold 2.4 min at 99.5% of methanol 50%, acetonitrile 50%, 0.05% PFPA, and 0.01% FA. A third aliquot was analyzed using basic negative ion-optimized conditions with a separate dedicated C18 column (LC neg). The basic extracts were gradient eluted from the column using methanol 95% and water 5%, with 6.5 mM ammonium bicarbonate at pH 8. Elution was performed at 0.35 mL min$^{-1}$ with a linear gradient from 0.5% to 70% of methanol 95%, water 5% with 6.5 mM ammonium bicarbonate over 4 min, followed by a rapid gradient to 99% in 0.5 min. The sample injection volume was 5 μL and a 2× needle loop overfill was used. Separations utilized separate acid and base-dedicated 2.1 × 100 mm Waters BEH C18 1.7 μm columns held at 40 °C. The fourth aliquot was analyzed via negative ionization following elution from an HILIC column (LC HILIC) (Waters UPLC BEH Amide 2.1 × 150 mm, 1.7 μm, held at 40 °C) using a gradient consisting of water (15%), methanol (5%), and acetonitrile (80%) with 10 mM ammonium formate, pH 10.16. Elution flow rate was 0.5 mL/min with a linear gradient from 5% to 50% in 3.5 min, followed by a linear gradient from 50% to 95% in 2 min, of water (50%), acetonitrile (50%) with 10 mM ammonium formate, pH 10.6. The MS analysis alternated between MS and data-dependent MS$^n$ scans using dynamic exclusion. The scan range varied slightly between methods but covered 70–1000 $m/z$.

**Quality assurance and quality control (QA/QC).** Several types of controls were analyzed in concert with the experimental samples: a pooled matrix sample generated by taking a small volume of each experimental sample (or alternatively, use of a pool of well-characterized human plasma, named MTRX for sample matrix) served as a technical replicate throughout the dataset; extracted water samples served as process blanks; and a cocktail of QC standards listed below, which were carefully chosen not to interfere with the measurement of endogenous compounds were spiked into every analyzed sample, allowed instrument performance monitoring, and aided chromatographic alignment. In LC neg conditions, internal standards were D7-glucose, d3-methionine, d3-leucine, d8-phenylalanine, d5-tryptophan, bromophenylalanine, d15-octanoic acid, d19-decanoic acid, d27-tetradecanoic acid, d35-octadecanoic acid, d2-eicosanoic acid. In LC HILIC conditions, internal standards were D35-octadecanoic acid, d5-indole acetic acid, bromophenylalanine, d5-tryptophan, d4-tyrosine, d3-serine, d3-aspartic acid, d7-ornithine, d4-lysine. In LC pos conditions, internal standards were d7-glucose, d3-methionine, d3-leucine, d8-phenylalanine, d5-tryptophan, bromophenylalanine, d4-tyrosine, d5-indole acetic acid, d5-hippuric acid, amitriptyline, d9-progesterone, d4-dioctylphthalate.

Instrument variability was determined by calculating the median relative standard deviation (RSD) for the internal standards that were added to each sample prior to injection into the mass spectrometers (median RSD = 3–4%). Instruments are calibrated at least weekly in the utilized polarity using thermo and mass accuracy is monitored at the batch level for the internal standards. A batch fails QC if any of the internal standards are more than 5 ppm away from the theoretical mass.

Overall process variability was determined by calculating the median RSD for all endogenous metabolites (i.e., non-instrument standards) present in 100% of the MTRX samples, which are technical replicates created from a large pool of extensively characterized human plasma. The median RSD for the MTRX samples was equal to 9–10%. Five MTRX samples and three process blank samples were processed per every batch of 30 samples. Experimental samples were randomized across the platform run with QC samples spaced evenly among the injections.

**Compounds identification and quantification.** Raw data were extracted, peak-identified, and QC processed using Metabolon's hardware and software. Compounds were identified by comparison to library entries of purified standards or recurrent unknown entities[59,60]. Briefly, Metabolon maintains a library based on authenticated standards that contains the retention time/index (RI), mass to charge ratio ($m/z$), and chromatographic data (including MS/MS spectral data) on all molecules present in the library. Furthermore, biochemical identifications are based on three criteria: retention index within a narrow RI window of the proposed identification, accurate mass match to the library ±10 ppm, and the MS/MS forward and reverse scores between the experimental data and authentic standards. The MS/MS scores are based on a comparison of the ions present in the experimental spectrum to the ions present in the library spectrum. While there may be similarities between these molecules based on one of these factors, the use of all three data points can be utilized to distinguish and differentiate biochemicals. More than 3300 commercially available purified standard compounds have been acquired and registered for analysis on all platforms for determination of their analytical characteristics. The full list of identified metabolites in both cohorts is available in Supplementary Data 16. Microbiota-derived metabolites identification was based on the Human Metabolome Database (www.hmdb.ca). The QC and curation processes were designed to ensure accurate and consistent identification of true chemical entities, and to remove those representing system artifacts, mis-assignments, and background noise. Metabolon data analysts use proprietary visualization and interpretation software to confirm the consistency of peak identification among the various samples. Library matches for each compound were checked for each sample and corrected if necessary. Peaks were quantified using area-under-the-curve. A data normalization step was performed to correct variation resulting from instrument inter-day tuning differences. Essentially, each compound was corrected in run-day blocks by registering the medians to equal one (1.00) and normalizing each data point proportionately

**Data analysis.** Metabolite amounts were analyzed using a three-step procedure, both for the comparison between recipients according to the GVHD status [recipient with acute GVHD, Ra vs those without GVH, Rs], and for the comparison between donors and their recipients without GVHD [D vs Rs] or with GVHD [D vs Ra]. The first two steps dealt with each metabolite separately, whereas the third and fourth step analyzed the whole dataset at once. These steps are detailed below; briefly, the two first steps differed in terms of outcomes that were compared across groups, either the proportion of non-detectable metabolite quantities (first step) or the average metabolite quantity (second step); the third step specifically aimed to detect groups of metabolites that experience similar pattern of changes across groups. For the Rs vs Ra comparison, a fourth step specifically aimed to detect metabolites that best predict the subject's group.

For D vs Rs and for D vs Ra, analyses comparisons were performed considering the donor–recipient pairs; consequently, only donors with a matched recipient who did not develop GVHD were considered. All metabolites were kept for the main analysis. As a sensitivity analysis, all donors were included, ignoring matching with their recipients, and led to similar conclusions.

Analyses for Ra vs Rs comparison were performed considering the two groups of patients independently. Only natural metabolites, including bacteria and

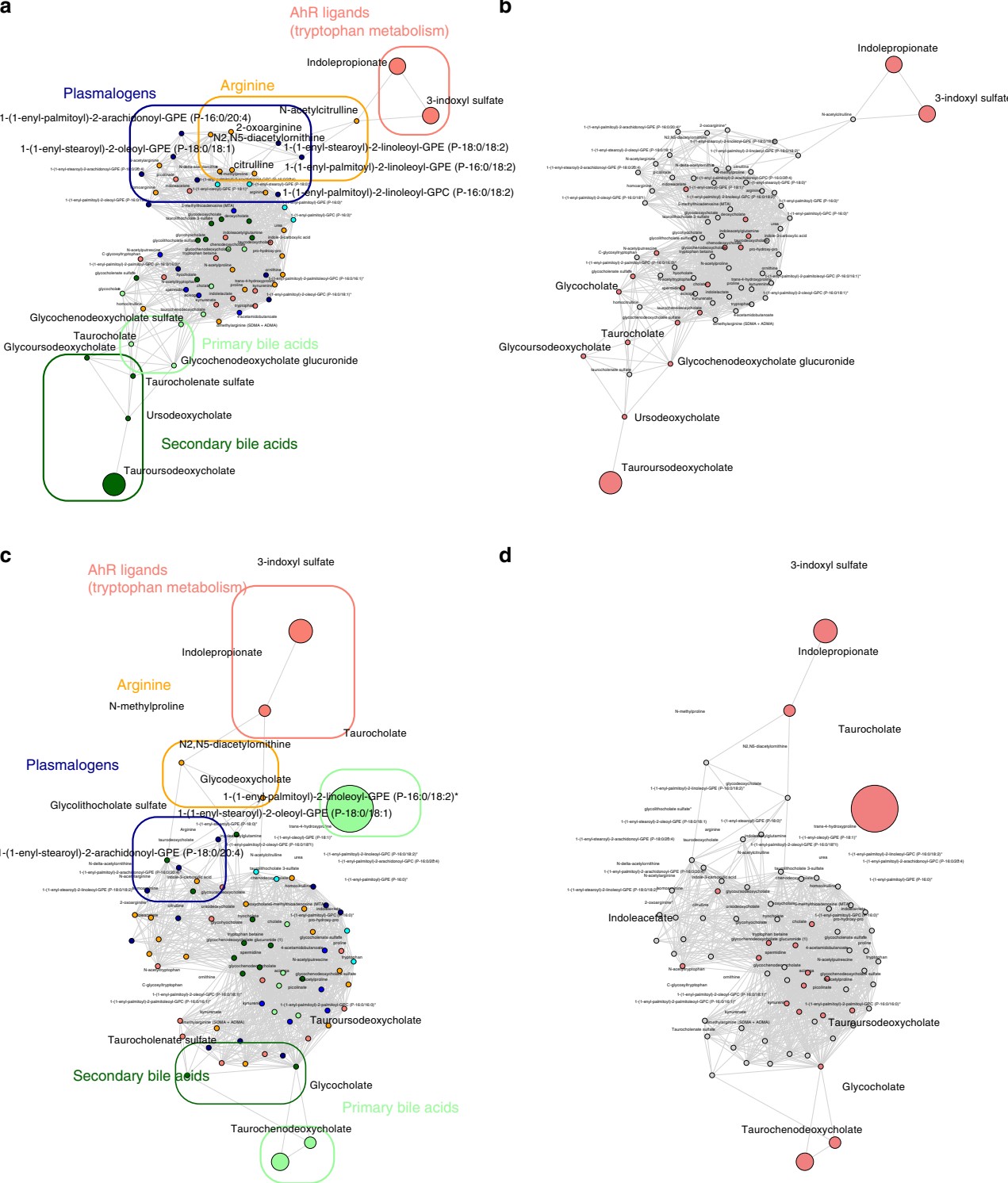

**Fig. 5 Metabolomics variation at aGvHD onset is not affected by age, gender, nor BMI.** Recipients with or without GvHD were compared after adjustment for age, gender, and body mass index (BMI) and re-analyzed to avoid any metabolite variation due to these parameters. The list of 57 metabolites significantly changing with acute GvHD onset in both cohorts is available in Supplementary Data 13. The main pathways involved at GvHD onset were used to build a network of compositional data where nodes represent metabolites and are linked if the ratio between corresponding metabolites was not significantly different between groups of patients. The same network was first colored according to the considered sub-pathway for cohorts 1 and 2 (**a** and **c**, respectively) and then according to microbial-derived metabolites (red nodes) (**b** and **d**, respectively).

fungi-derived metabolites, were considered, excluding drugs or assimilated compounds. A sensitivity analysis was done based on a 1:1 matching on sex and age between receivers with (Ra, cases) or without (Rs, controls) GVHD, which led to similar results. This pairing was done by matching each Ra patient with the Rs patient with the closest age and sex. After pairing, there was four gender mismatch

in the Saint Louis' cohort and nine in the Cryostem's cohort. The age difference in a given pair was between −3 and +1 year in the Saint Louis' cohort and −11 and +6 years in the Cryostem one.

All analyses were performed first on the Saint Louis hospital cohort (exploratory analysis), then on the Cryostem cohort (confirmatory analysis). The

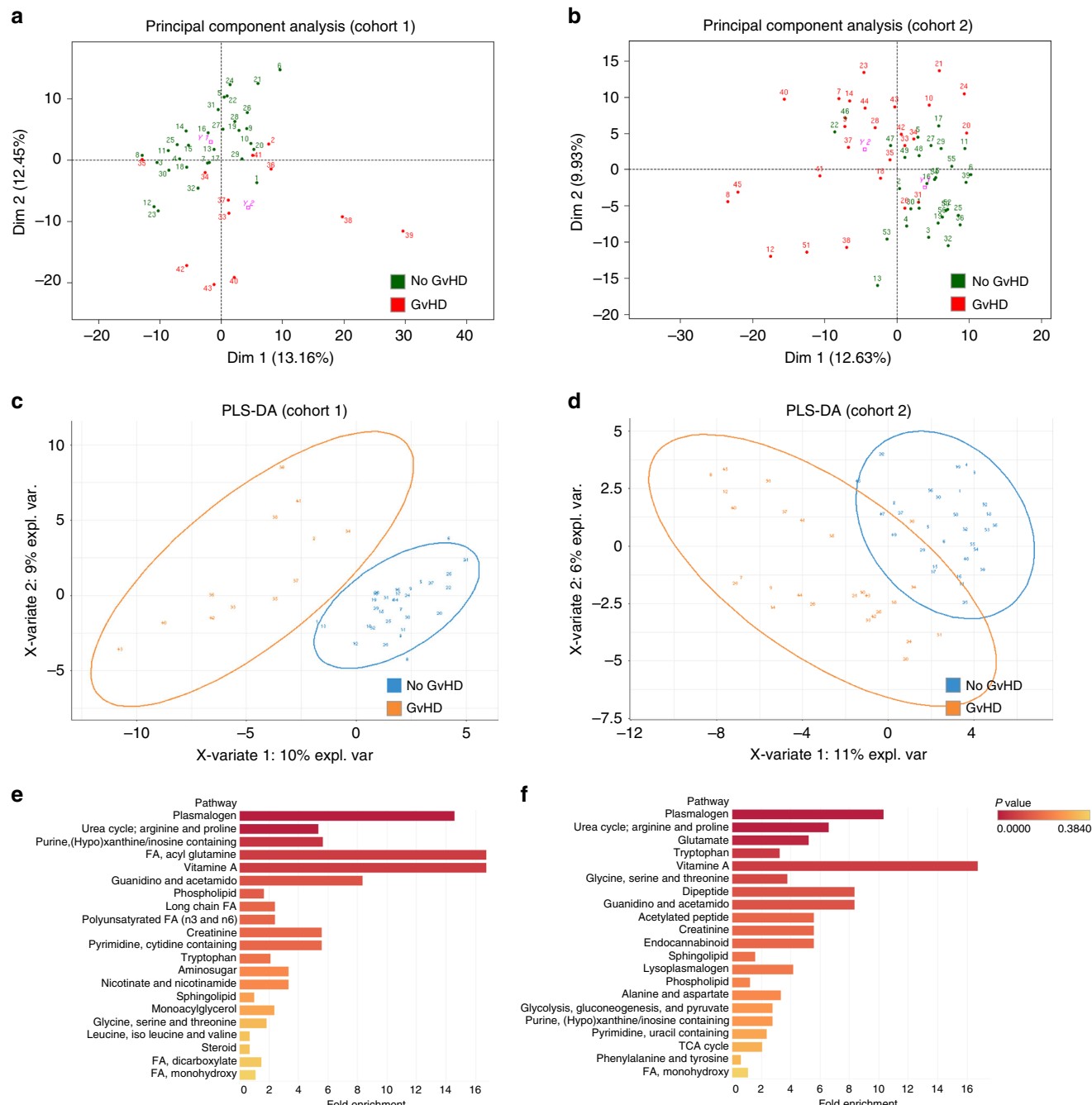

**Fig. 6 Identification of main pathways involved in aGVHD using multivariate analysis.** Principal component analysis was able to discriminate recipient with (red) or without (green) GvHD in dimension 1 for cohort 1 (**a**) and 2 (**b**). Sparse partial least square discriminant analysis (sPLS-DA) was then used to identify metabolites that mostly contribute to the discrimination of patients with (orange) or without GvHD (blue) for cohort 1 (**c**) and 2 (**d**). Complete list of metabolites is available in Supplementary Data 14. These metabolites were then used to identify main pathways involved in acute GvHD using overrepresentation analysis for cohort 1 (**e**) and 2 (**f**).

confirmatory analysis used the same filters and procedures than the exploratory analysis to assess whether the results obtained in the first cohort of patients were consistent in an independent cohort. To ensure the comparability of the results, both analyses used only the 653 metabolites that were shared by the two datasets (excluding 150 metabolites detected only in the Saint Louis cohort and 274 metabolites detected only in the Cryostem cohort).

Unless specified, all analyses were done using a 5% type I error rate ($p < 0.05$, after multiplicity correction when required). To handle obvious confounders, all analyses of steps two and three (detailed below) were secondly adjusted. More specifically, unpaired analyses were adjusted for age, sex, and BMI. Analyses on paired cases (either D vs Rs, or D vs Ra) were adjusted for differences of age and of BMI, as well as for sex match between the two subjects of the pair. For paired

analysis of Ra vs paired Rs, given matching was performed on age and sex, estimates were only adjusted for difference in BMI between the two subjects of a pair.

All analyses were performed using R and appropriate additional packages, as described below.

The first step of analysis consisted in selection of metabolites and imputation of undetectable amounts. Since some metabolites could not be detected in several patients, indicating low amounts of metabolites in these patients, we first wondered whether these non-detectable amounts occurred randomly or preferentially in one of the two groups. This was done, for each metabolite, by replacing the values by 1 if an amount could be detected (that is, a numerical value was present in the dataset) and 0 otherwise (that is, the value was missing in the dataset). The proportion of one was then compared between the two groups. For paired data, the

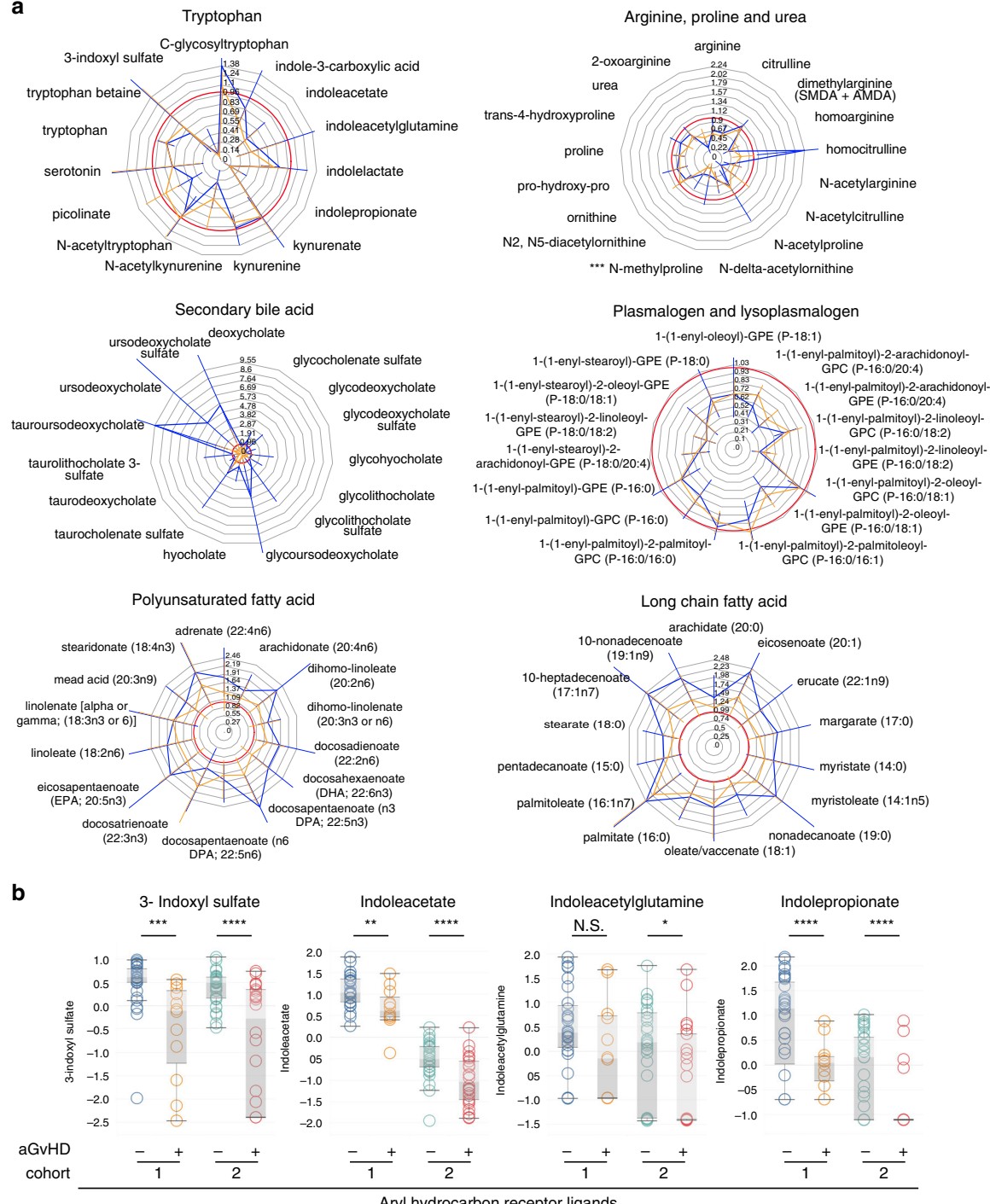

**Fig. 7 Pathways associated with aGVHD involve microbiota-derived aryl hydrocarbon receptor ligands.** Metabolites from the six most frequently involved sub-pathways at GvHD onset were compared between recipient with and without GvHD in both cohorts (blue: monocentric cohort 1; orange: multicentric cohort 2; red line: ratio = 1). Radar plots represent the ratio of the mean value of each metabolite in recipients with GvHD compared to those without GvHD, with bars corresponding to the standard deviation of the mean value in both cohorts. The red line corresponds to a ratio of 1. **b** Within the tryptophan pathway, the four metabolites that were decreased at GvHD onset were microbial-derived indole compounds that may act as aryl hydrocarbon receptor (AhR) ligands. Logarithm of the individual values are represented for both cohorts in patients with or without GvHD and values were compared with a Student test. Each box represents the median value with interquartile range, whiskers described minimal and maximal values. *$p < 0.05$, **$p < 0.01$, ***$p < 0.001$, ****$p < 0.0001$.

comparison was done using the exact version of the McNemar's test (that is, binomial test to compare the proportion of discordant pairs in one direction to 0.5). For unpaired data, the comparison was done using the exact Fisher's test for 2 × 2 contingency tables.

A metabolite was declared to be significantly more frequently undetectable in one group (hence, present at significantly lower amounts in patients of that group) when the test was significant after Bonferroni correction for multiple testing. For the D vs Rs analysis, all the 653 metabolites were analyzed, leading

to significant differences for $p < 7.66 \times 10^{-5}$. For the Rs vs Ra analysis, 524 metabolites were analyzed, leading to significant results when $p < 9.54 \times 10^{-5}$.

After this first step, chemicals that could not be detected in at least one half of the patients either overall or in either group of the Saint Louis hospital cohort were excluded for the following two steps. As stated above, the resulting list of excluded chemicals also applied to the Cryostem cohort dataset.

Based on the experimental protocol, missing data can be considered mainly to rely on the limit of quantification (LOQ) of the analytical method: any value below this limit could not be measured and reported (BLQ data). To consider the information of such BLQ data—known as left-censoring—the most usual method is to impute half the LoQ to missing data. Since LoQ is unknown for each chemical, imputations used half the minimal observed value. It allows unbiased estimates unless a large amount of data is missing (but such cases were removed by filtering at the end of step 1 of the analysis). These findings were notably reported in Keizer et al.[61], who showed that except when the percentage of missing data is high, imputing values by LOQ/2 achieves less bias than simply discarding the data, and in fact is similar to having complete data or using more complex methods than the ones used in this paper. Its main drawback is to generate ties in the sample, thus resulting in impaired estimates of variances. Therefore, a small random noise generated from a Gaussian distribution was added to imputed values, and then rounded to the nearest integer to maintain the characteristics of the original data. A 100 variance value of the Gaussian distribution was used to ensure a lower variability than that observed in detected amounts. To check that this additional step did not introduced bias, all analyses were also performed without this additional noise. Similarly, to check the assumption that using LOQ/2 does not introduces a bias, analyses were also performed using other fractions of the minimal value (25%, 75%, 90%, 95%, 99%) or the minimal value itself, another common approach[62]. Because of the limited sample size, and of the lack of sensitivity of the results to these choices, more complex imputation methods like fitting a truncated distribution to the data were not tested.

At the end of this step, a principal component analysis (PCA) and a sparse partial least square discriminant analysis (sPLS-DA) were performed to check that the different groups were indeed separated, allowing identifying metabolites whose level differ between the groups (R package FactoMineR[63], function PCA and R package mixOmics[64], function plsda and splsda). Due to the small sample size and the very low (sample number)/(predictor number) ratio, PLS-DA is prone to overfitting and cross-validation cannot be used here. Hence, PLS-DA results should be taken as descriptive.

Metabolites that were identified in sPLS-DA were then used to build an ORA. Enrichment ($E$) was calculated by considering the number of metabolites identified with sPLS-DA in each pathways ($k$), the total number of metabolites identified in sPLSD-DA ($n$), the number of metabolites in each pathway ($m$), the total number of metabolites used for analysis ($M$) as follows: $E = (k/m)/((n-k)/(N-m))$. For each pathway, $p$ value was determined by calculation of the hypergeometric distribution.

The second step of analysis aimed to compare average amount for each compound. After filtering and imputation, each chemical was analyzed separately. For a given metabolite, the amounts between the two groups were compared, after log transformation, using two-sided Student's $T$ test, with Aspin–Welch correction for unequal variances in the case of unpaired data; the paired version of the test was used for paired data. A chemical was declared to be present in significantly different amounts in the two groups if the test was significant after Bonferroni's correction for multiple testing.

For the D vs Rs analysis, 83 metabolites were excluded by filtering on non-detectable values, leading to 570 metabolites for further analyses. Consequently, $p < 8.77 \times 10^{-5}$ (that is, $p^\star < 0.05$ where $p^\star$ is $p$ after Bonferroni correction) was used as the threshold to detect significantly differentially present metabolites.

For the Rs vs Ra analysis, 35 metabolites were excluded by filtering on non-detectable values, leading to 491 metabolites to analyze. Consequently, $p < 1.02 \times 10^{-4}$ (that is, $p^\star < 0.05$ where $p^\star$ is $p$ after Bonferroni correction) was used as the threshold to detect significantly differentially present metabolites.

Heatmap was constructed on metabolites identified with the two-sided Student's $T$-test in the previous step. Hierarchical clustering analysis of data was performed using Euclidian distance measurement of similarity followed by a clustering algorithm based on Ward's linkage, that cluster data to minimize the sum of squares of any two clusters. Analyses were performed using the hclust function from R v3.5.1.

The third step of analysis aimed to identify similarly behaving chemicals. The global comparison was based on the idea that, rather than absolute amounts, the relative amounts between chemicals are informative—first, because biologically they better represent the equilibria between different chemicals, hence the relative importance of different pathways and secondly, because using ratios lowers the necessity of a perfect normalization of the data. Consequently, the change in the ratio of two metabolites between the two groups was the parameter of interest.

Since the number of ratios increases as the square of the number of tested metabolites, and since these ratios are obviously correlated, analyzing all ratios individually with multiple testing corrections would be very inefficient (that is, achieving a very low power). We used instead a global comparison of all pairwise ratios, according to the method described in the paper by Curis et al.[65] and briefly described thereafter.

Basically, the idea is to build an undirected graph (network) where each node is a chemical and two nodes are connected if, and only if, their ratio is unchanged between the two groups. The theoretical graph is a set of disjoint subgraphs, each subgraph being a clique (that is, in this subgraph, each node is connected to all other nodes). In a given clique, all chemicals experience no change in their relative amount, hence their absolute amount changes identically (eventually, does no change) between the two groups. Conversely, chemicals belonging to different cliques experience a change in their relative amount, hence different changes in their absolute amount, with at least one of them being modified between the two groups.

To build the observed graph, the ratio is tested and if the $p$ value of the test is below a predefined threshold, $p_{\max}$, the connection between the two corresponding is removed (the ratio has changed between the two groups). This is done for all possible ratios, and the resulting graph is then analyzed. If it contains more than one subgraph, then at most one of these subgraphs correspond to "no difference between the groups", and all other subgraphs contain chemicals that do change between the two groups. The largest subgraph will be taken as the "no change" subgraph.

The key step is the choice of the threshold, $p_{\max}$. Obviously, the larger the threshold is, the more likely is the discovery of unconnected subgraphs. Hence, $p_{\max}$ was selected to minimize the probability of observing at least two disjoint subgraphs, under the null hypothesis that no chemical changes (in graph terms, that the theoretical graph contains a single clique). This was achieved by simulation: under the null hypothesis and assuming independent chemicals, 10,000 simulations were done with the same sample size and the same number of nodes than in our datasets, using a log-normal distribution; $p_{\max}$ was estimated as the threshold to use so that the proportion of simulations that give two or more disconnected subgraphs is at most 5%. Practically, to account for simulation uncertainties and deviations to the simulation model, we used the lowest bound of the 95% confidence interval of $p_{\max}$ obtained by simulation, truncated to the second decimal, ensuring a conservative test. The Supplementary Data 17 gives the threshold obtained and used for our different datasets in main analyses.

This third step was performed twice: first on all metabolites (after exclusion of metabolites with too large non-detectable amounts, as reported above), and secondly only on metabolites from a preselected set of candidate metabolic pathways of interest: polyamine metabolism, tryptophan metabolism, urea cycle, arginine & proline metabolism, bacterial or fungal, plasmalogen or lysoplasmalogen, and primary or secondary BA metabolism.

All the analyses for this third step, including simulations, were done using the SARP.compo package for R[65], available on the CRAN repository.

Last, the fourth step selected metabolites using group membership modeling. One may wonder whether some of the metabolites selected as associated with either group, may have been selected only due to their relationships with the others. Therefore, multivariable analyses were performed, considering (i) the large number of metabolites (491 for the Rs vs Ra comparison) against the small number of observations (overall, 43 in the Saint Louis sample and 56 in the Cryostem sample) and (ii) the low prevalence of GvHD, notably in the Saint Louis sample (with only 12 GvHD). The latter issue avoids any multivariable regression in the Saint Louis sample (indeed, it is commonly reported that at least 10 events should be observed when including one variable in the model). Then, to handle the first issue, lasso logistic regression—following its original use in linear models[66]—was used. It tends to produce some model coefficients that are exactly 0 and hence gives interpretable models. Thus, this technic is robust in the context of our analysis, due to its tendency to prefer solutions with fewer non-zero coefficients, effectively reducing the number of features upon which the given solution is dependent

**Reporting summary**. Further information on research design is available in the Nature Research Reporting Summary linked to this article.

## Data availability

The source data underlying Fig. 1c are provided as a Source Data file. Raw data underlying Figs. 2–7 are available on MetaboLights repository for both cohorts: MTBLS204 (cohort 1) and MTBLS205 (cohort 2). All other data are available from the corresponding author on reasonable request.

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

## Acknowledgements

The authors thank all members of the CRYOSTEM Consortium and of the Francophone Society of Marrow Transplantations and Cellular Therapy (SFGM-TC) for providing patients samples used in this study and for their support: University Hospital of Angers, University Hospital of Dijon Bourgogne, University Hospital of Besançon, University Hospital of Grenoble, University Hospital of Lille, University Hospital of Lyon, University Hospital of Bordeaux, Paoli-Calmettes Institute, AP-HM, University Hospital of Nantes, AP-HP (Saint Louis Hospital, La Pitié-Salpêtrière Hospital, Saint-Antoine Hospital, Robert Debré Hospital, Necker Hospital, Groupe Hospitalier Henri Mondor), INSERM, University Hospital of Toulouse, University Hospital of Tours, University Hospital of Rennes, University Hospital of Clermont-Ferrand, University Hospital of Saint-Etienne, Lucien Neuwirth Cancer Institute, University Hospital of Poitiers, University Hospital of Nice, University Hospital of Brest, Military Hospital of Percy, University Hospital of Montpellier, Gustave Roussy Institute, University Hospital of Limoges, University Hospital of Caen, Etablissement Français du Sang. We thank Alexion Company for their financial support. This project was funded by a grant from the Direction Générale de l'Offre de Soins (DGOS) and the Agence National de la Recherche (ANR) within a Projet de Recherche Translationnelle en Santé (ANR-PRTS 13002). G.S. received a research grant from Alexion Pharmaceutical company (APALEX) and from the Institut National du Cancer (project number INCa 2014-1-PL BIO-07-IP-1). CRYOSTEM project is supported by a grant from the Institut National du Cancer (INCa) and the Agence Nationale de la Recherche (ANR) and under the auspices of the Société Francophone de Greffe de Moelle et de Thérapie Cellulaire (SFGM-TC).

## Author contributions

D.M. and E.L. performed the experiments, collected and analyzed the data, designed figures. E.C and S.C. performed statistical analysis. L.D., S.R., and B.I. performed the experiments. R.P.L. coordinated Cryostem consortium and provided access to samples from cohort 2. M.R. and F.S.F. collected clinical data. D.M. and G.S wrote the manuscript. All authors provided feedback on the manuscript. D.M., S.C., L.R., and G.S. supervised and conceived the project. R.P.L. and G.S. secured funding for the study.

## Competing interests

The authors declare no competing interests.
