## [Peer Review File · Nature Communications]

Reviewers' comments:

Reviewer #1 (Remarks to the Author):

This is a novel exploration utilizing cutting edge technologies for assessing the metabolomic effects of GVHD in humans. The authors have generally done as good a job as can be done for these types of retrospective analysis. The limitations remain from such analysis, which the authors should address in their discussions. Additional issues include:

1. The sample collections for the analysis between the GVHD+ and -ve groups are not matched for time or immunosuppression. The authors should discuss the relevance/ impact of this.
2. The comparison of metabolomes from donors in MRD patients will need better clarification. The manuscript, perhaps unintentionally implies that it reflects the 'immune metabolic' of donor cells. Clearly circulating metabolites in the donors and recipients are reflection of many biological processes-diet-microbiome, and information on these aspects is not provided or available from the authors. The authors should acknowledge these limitations.
3. Please clarify whether the non-GVHD cohort never developed late-onset acute GVHD.
4. The utilization of antibiotics in these cohorts of patients, if available, should be presented and discussed given impact on microbiome. If unavailable, it should be discussed as a caveat in interpreting the results.

Reviewer #2 (Remarks to the Author):

In their paper, "Metabolomics in human acute GvH Disease", Michonneau et al. attempt to understand the metabolomic factors underlying GvH disease in recipients of allogeneic hematopoietic stem cell transplantation.

The fact that all of the metabolites are linked together in the system is highlighted by the authors in multiple places in the paper. However, in the statistical analysis, the authors perform multiple hypothesis testing in a univariate fashion, which explicitly assumes that each metabolite is independent of others in the system. Thus, the statistical analysis in the paper is not in keeping with the initial hypothesis proposed by the authors. It is more appropriate to

to approach this analysis from a systems perspective and run a multivariable regression to detect differences in the metabolites while accounting for all the other metabolites in the same samples.

....

Page 2, Line 75: In epidemiology, controls are a study group who are likely to acquire the disease and have the same risk factors as the cases - who end up getting the disease. Calling the sibling donors "controls" is a confusing use of the term. The samples that the study team acquired from the transplant recipient who did not acquire GvHD at day 90 can be labeled as controls.

Figure 2a:

Using the word, "Filtration" is ambiguous. Please reword this sentence: Filtration of 83 metabolites ...: as "Discarded 83 metabolites ..."

Page 2, Line 92:

It is misleading to state that 37 and 45 metabolites were different between GvHD recipients and their donors in cohort 1 and 2. In Figure 2a, the statistical test with correction for false positives shows that only 5 were different between GvHD recipients and their donors in cohort 1. The corresponding number for cohort 2 is unclear because the tables in the supplementary tables are not correctly formatted. Please see suggestions for correcting this format below.

Presentation of Supplementary Tables

The supplementary tables are unreadable. Each table is split across two pages that are not sequential. Would the authors please reformat in landscape format on a larger area (i.e. A3 size or higher in PDF format) so that each row of a table is at least captured in a single page?

Figure 3:

Parts a + b show that none of the metabolites are significantly different after correction for false positives after multiple hypothesis testing. In the text, Page 3, Line 122-125 state that a number of metabolites are "significantly different after Bonferroni correction". The figure does not show this finding. These results are thus inconsistent with the findings shown in the figure.

Under Data Analysis: Page 6, Line 268-269:

"The confirmatory analysis used the same filters and procedures than the exploratory analysis to ensure a true validation analysis." A true validation analysis would take the significantly different metabolites from the exploration stage and use these metabolites to predict disease state in an independent cohort. If the metabolites that were detected as different in the exploratory state were also found to be predictive of the disease state, then true validation of the metabolites association with disease could be claimed. In the absence of such an analyses, all that the authors have attempted to do is to test whether their findings were consistent across two independent cohorts. Recommend removing the phrase "true validation analysis". Alternatively, the two cohorts could have been combined in a single multivariable regression analysis with an indicator variable for cohort to test consistency and increase the statistical power of the analyses. Using multivariable analyses would be consistent with modern statistical approaches in the field.

Imputation Strategy: The method of imputation with a fixed value followed by addition of Gaussian noise introduces directional bias into the analyses. Have the authors considered a more robust form of imputation such as multiple imputation? Please see methods here:

<https://www.ncbi.nlm.nih.gov/pmc/articles/PMC3074241/>

Description of Study population

What are the clinical risk factors for GvHD outside of metabolites? Were these clinical risk factors equivalently represented in the transplant recipients with and without GvHD? Please present a table comparing the basic demographics of recipients in the two cohorts, the donors and a table comparing the demographics of recipients who acquired GvHD vs. recipients who did not acquire GvHD. Please include a comparison of clinical factors that were collected as part of the study (Page 5, Line 188 through Line 191)

Minor Comments

Line 82: "The distribution of these metabolites ..." Which set of metabolites are the authors referring to - the shared ones or across all metabolites? Please clarify. If shared, modify the sentence as : "The distribution of these shared metabolites ..."

Lines 82-84: Do the percentages indicate average percentages or median percentages? Please clarify.

Page 2, Line 99: "fatty acid, mono and diacylglycerol and primary bile acid (BA)": do these correspond to the 5 metabolites that are shown as significantly different after correction for false positives after univariate multiple hypothesis testing?

Reviewer #3 (Remarks to the Author):

This is a paper which has explored the metabolic changes of graft versus host disease (GvHD) after allogeneic stem cell transplantation both in comparison to the original donors and in comparison to a transplant group that did not experience GvHD. There are two independent cohorts examined in this study which should allow for more robust conclusions to be drawn from the results. In general, I believe the paper will be of interest to the medical community, and its potential insights into the effects of graft versus host disease may be of interest to the wider immunological research community.

I have some minor concerns about the data analysis that has been conducted and some major concerns about the reporting of the metabolomics experiments.

General comments

Figures: Much of the figure text is too small to read comfortably and should be increased to size 12 Arial or equivalent.

Tables: All of the tables featuring metabolite data fail to report important information on the average detected intensity for each metabolite per class and the range per class. Standard deviation should also be reported.

Statistics:

You have compared the D v R response and the Ra v. Rs response but nowhere have you reported the results of the D v R comparison for the two classes of recipient. I think the latter is an important comparison given the general variability in human data – the use of sibling pairs will remove some of this effect and may prove insightful. I would have anticipated some assessment of confounding factors (age and BMI being the two major possibilities) with some statistical approach used to account for this. An attempt to do this retrospectively by pairing samples was done but no details were given on how well matched these pairs were. BMI could play an important role in some of the results, especially as it is known to be correlated with both lipid levels and the faecal microbiome.

You have assessed absolute differences in missing values between the sample classes but have not assessed whether this missingness could in fact be related to the average detected intensity of the metabolite in question. If these are poorly detected metabolites, you may be assessing only that

they are barely reaching the limit of detection. Equally, you could be measuring poorly controlled storage and collection conditions – as the protocols for these were not given, it cannot be judged.

You have removed data for which 50% of values are missing, but have used half minimum values to substitute the remainder of missing values (MV). This will potentially severely bias your results, especially when there appears to have been no assessment on whether the missing values were correlated with intensity or not. I would strongly recommend consulting an independent statistician for guidance on this. Even if the approach was justified, imputation may not be required for univariate statistical methods and it will bias your results less to type 1 errors if it is avoided.

Methods

You have not followed good reporting practice for metabolomics methods as laid out in Sumner, L.W., Amberg, A., Barrett, D. et al. *Metabolomics* (2007) 3: 211. <https://doi.org/10.1007/s11306-007-0082-2> and Goodacre, R., Broadhurst, D., Smilde, A.K. et al. *Metabolomics* (2007) 3: 231. <https://doi.org/10.1007/s11306-007-0081-3>. The details provided in your methods are not suitable for someone else to be able to replicate your experiment to any degree and thus fall short of the requirements of good scientific reporting.

Specific comments:

Table 1: There is a major confounding factor of the average day at which blood was drawn from Ra (<23 or 28 days) vs Rs (90 days) – this was not highlighted or accounted for in the statistics or discussion.

There are no statistics on the differences between the donors and recipients for BMI

There are no statistics on the difference of Ra vs Rs groups given for either cohort.

Given the importance attached to the tryptophan pathway, there is no discussion on potentially different diets between groups, especially dietary fiber intake. This may not have been collected at the time, but I would have anticipated a discussion on it as a confounding factor in the results.

Line 80 There is a mismatch between the figures reported here and the figures reported in Fig 1b. Why?

Figure 1c: Illegible

Supp figures 3 and 5 describe themselves as cohorts 1 and 2 from St Louis. Is this a mistake or were there two cohorts from St Louis?

Line 104 „BA, tryptophan and polyamine metabolites were the pathways most affected (Fig 2 and supp Fig 1). The polyamines do not seem from the figures given here to be unduly affected and there is only one metabolite from the tryptophan pathway listed. The supplementary tables give more evidence to this statement.

Figure 4a: text too small and no error bars provided on radar plots.

Figure 4b: the ligands are not differentiated. Indolepropionate is also not a dose responsive ligand of AHRs (e.g. see Hubbard et al 2015).

Line 204: protocol fails to give anticoagulant, time or temperature of processing

Line 209: how much plasma was eventually extracted and what were the recovery standards used and at what concentrations?

Line 222 which solvents were the samples reconstituted with?

Line 223 which standards were used and at what concentrations?

Line 218 to 236 please give exact details of gradients and conditions used as is concordant with standard reporting practices for UPLC-MS

Line 235 please give details of data dependent conditions

Line 238: which ions were used for identification and what quality controls were conducted to ensure they were accurate, especially with regard to retention time matching to a library. How many ions were used as the minimum qualifying ions for identification

Line 250: please give details of what quality controls and quality assurance you used to ensure the mass spectrometry was robust. How many quality control samples and what was the median standard deviation of these quality control samples throughout the run?

How was the run randomised?

What was the exact data processing method used – normalisation, peak picking, noise reduction, filtering, software used?

Were metabolites exactly quantified or only relatively quantified. To which standards?

Line 263: you have excluded both drugs and nutrients (is tryptophan not a nutrient???) but both are important to the eventual findings and should have been tested as possibly influencing factors on the results.

Line 276-288: this seems a very simplistic approach to me. Results should have been at least correlated with intensities, or weighted in the scoring based on the average intensity.

Line 293: see earlier comment about MV imputation

Line 312: metabolites with a high variance may have large mean fold changes that are meaningless when they are not robust in their measurement e.g. high technical or natural biological variance. Identifying high fold changes alone may just be adding noise to resulting interpretation.

Responses to the Reviewers' comments & queries:

We thank the reviewers for their useful comments that in our view greatly improved the manuscript. Enclosed are point by point answers to the reviewers' questions and queries. Modifications in responses to these queries are included in this document in *blue italic* font and highlighted in yellow in the revised manuscript.

Reviewer #1:

This is a novel exploration utilizing cutting edge technologies for assessing the metabolomic effects of GVHD in humans. The authors have generally done as good a job as can be done for these types of retrospective analysis. The limitations remain from such analysis, which the authors should address in their discussions. Additional issues include:

1. The sample collections for the analysis between the GVHD+ and -ve groups are not matched for time or immunosuppression. The authors should discuss the relevance/ impact of this.

This is a very important point we added to discussion. When we designed the research project we took into account the feasibility of the study both at the single center level (Hospital St Louis) and in the confirmatory cohort (Cryostem) where patients were treated in different French centers. We had to define a timepoint for patients without GvHD that was, somewhat arbitrary, defined as day + 90 after transplantation. Patients with GvHD were sampled at GvHD onset, before corticosteroid treatments. This resulted in slight variations between transplantation and sampling (mean day+40 in cohort 1 and day+33 in cohort 2, median delay day+26 in cohort 1 and day+28 in cohort 2). Even if we cannot exclude that this difference could have contributed to some variations observed between patients with or without GvHD, the impact of difference could be minored by the fact that almost all patient had natural oral feeding, similar antibiotics treatments (see point 4) and received the same immunosuppressive drugs at the time of sampling. Furthermore, all samples were obtained before corticosteroid treatment and all patients (with or without GvHD) received GvHD prophylaxis based on calcineurin inhibitors at the time of sampling, (consisting in cyclosporine in more than 90% of patients).

The following sentence was added in the discussion:

“Graft versus host disease is an immune reaction from donor immune cells targeting allogeneic antigens in recipients, whose pathophysiology is still only partially understood in Humans. Recent experimental researches have highlighted the complex network of interactions and regulations between immune cells, microbiota and host environment⁴. Here we report that patients who underwent allogeneic HSCT have major metabolomics changes compared to healthy subjects, and that acute GvHD onset seems to be associated with specific differences in metabolomics profile by comparison with patients who did not developed GvHD. Comparing patients without GvHD with those who developed an unpredictable time onset GvHD, leads to intrinsic differences between groups of patients that may have contributed to the observed variations in metabolites. These differences include: slight differences in the day of sampling after transplantation, feeding or treatments received at the time of sampling. To minimize the putative impact of these confounding factors, we used two independent cohorts of patients to confirm our results and explored groups of as

much as possible similar patient characteristics in terms of HLA-matching, immunosuppression or diet mode.”

2. The comparison of metabolomes from donors in MRD patients will need better clarification. The manuscript, perhaps unintentionally implies that it reflects the 'immune metabolic' of donor cells. Clearly circulating metabolites in the donors and recipients are reflection of many biological processes/diet/microbiome, and information on these aspects is not provided or available from the authors. The authors should acknowledge these limitations.

We do agree with Reviewer 1. Our aim was not to suggest that metabolome in recipients reflects donor cell metabolism. Metabolome reflects both host and microbiota metabolism. Metabolites detected in the recipient are mainly generated by recipient cells and microbiota metabolism. Even if donor hematopoietic cells could contribute to metabolome, it is clearly not the main contributing cells to circulating metabolites.

The following sentence was added to the text: *“In patients, metabolome mainly reflects the metabolism of host cells and of its microbiota.”*

3. Please clarify whether the non-GVHD cohort never developed late-onset acute GVHD.

As we retrospectively analyzed frozen samples from transplanted patients, we were able to consider only patients who NEVER developed neither classical nor late-onset acute GvHD in the non-GVHD groups. This point was clarified by adding the following sentence in the “patients and methods” section: *“Recipients’ samples were collected at day 90 ± 5 after transplantation for patients who did not develop acute GvHD at any time after transplantation, and at the onset of symptoms, before starting any corticosteroid treatment, for acute GvHD. Patients in the non-GVHD group never developed neither classical nor late onset acute GvHD”.*

4. The utilization of antibiotics in these cohorts of patients, if available, should be presented and discussed given impact on microbiome. If unavailable, it should be discussed as a caveat in interpreting the results.

These data are fully available for cohort 1 and were added to **supplementary table 1**. All patients received antibiotics (either as prophylaxis or for treatment) at the time of sampling. For cohort 2, we did not have access to this information in the clinical database, BUT antibiotics were detected with mass spectrometry in most patients. This, as reported by others, could be a reason why recipients’ microbiota is altered after transplantation and why multiple microbiota-derived metabolites are modified after transplantation and in GvHD patients (Jenq et al. BBMT, 2015). Alternatively, GvHD itself could also generate dysbiosis that could be involved in GvHD pathogenesis (Jenq et al. JEM 2012). These references are added to the manuscript and these points are now added to discussion:

“An unsolved question raised by these results is the relation between acute GvHD and metabolome alterations at disease onset. It is tempting to consider that at least a part of metabolites changes could be involved in GvHD pathogenesis, as suggested by recent results in animal model^{16,17}. Nevertheless, one cannot exclude that GvHD by itself could induce alteration of tissues metabolism. In fact, it has been previously demonstrated, mostly in experimental models, that dysbiosis observed in GvHD could be both partly causative and consequence of the allogeneic immune response^{7,16}. Our results suggest that many variations observed in metabolites could be due to microbiota changes after transplantation

especially if GvHD occur. In both cohorts, all patients received antibiotics at the time of sampling. Antibiotics are involved in dysbiosis of microbiota and most likely dysbiosis participates to metabolomics changes after transplantation. However, whether this phenomenon could be a factor increased acute GvHD severity remains to be explored. Interestingly, recent clinical studies suggested that antibiotics may have a major impact on GvHD-risk in Humans^{8,10}."

Reviewer #2:

In their paper, "Metabolomics in human acute GvH Disease", Michonneau et al. attempt to understand the metabolomic factors underlying GvH disease in recipients of allogeneic hematopoietic stem cell transplantation.

1- The fact that all of the metabolites are linked together in the system is highlighted by the authors in multiple places in the paper. However, in the statistical analysis, the authors perform multiple hypotheses testing in a univariate fashion, which explicitly assumes that each metabolite is independent of others in the system. Thus, the statistical analysis in the paper is not in keeping with the initial hypothesis proposed by the authors. It is more appropriate to approach this analysis from a systems perspective and run a multivariable regression to detect differences in the metabolites while accounting for all the other metabolites in the same samples.

We agree with the Reviewer that we only performed analyses to test for differences in distribution of each metabolite between diseased patients and healthy donors on one hand, or between acute GvHD patients and patients free of GvHD on the other hand, ignoring potential correlations between those metabolites. Thus, it is obvious that some of the metabolites selected as associated with either group, may have been selected only due to their relationships with the others. No "classical" multivariable analyses were performed, due to (i) the large number of metabolites (491 measurements) against the small number of observations (overall, 43 in the Saint Louis sample and 56 in the Cryostem sample), and (ii) the low prevalence of GvHD, notably in the Saint Louis cohort (with only 12 GvHD). The latter issue avoids any multivariable regression in the Saint Louis sample (indeed, it is commonly reported that at least 10 events should be observed when including one variable in the model). To handle the issue raised by the reviewer, lasso logistic regression - following its original use in linear models (Tibshirani 1996) - could be used. It tends to produce model coefficients that are exactly 0 and hence gives interpretable models. Thus, this technic is robust in this context due to its tendency to prefer solutions with fewer non-zero coefficients, effectively reducing the number of features upon which the given solution is dependent. This has been run with results tabulated below. Expectedly, only a few metabolites were selected, including 8 previously selected in univariate analyses.

The following text was added to results together with **table 1**: *"Considering the fact that metabolites variation might result from their interdependency, lasso logistic regression was performed to identify the set of metabolites that seems to be predictive of acute GvHD (table 1). Using lasso regression analysis, we were able to confirm that both tryptophan and arginine pathways were involved, especially the AhR ligand 3-indoxyl sulfate that was significantly detected in both cohorts."*

Furthermore, the following text was added to the methods section:

“Fourth step: metabolites predicting subjects’ group

Last, one may wonder whether some of the metabolites selected as associated with either group, may have been selected only due to their relationships with the others. Therefore, multivariable analyses were performed, considering (i) the large number of metabolites (491 for the Rs vs Ra comparison) against the small number of observations (overall, 43 in the Saint Louis sample and 56 in the Cryostem sample), and (ii) the low prevalence of GvHD, notably in the Saint Louis sample (with only 12 GvHD). The latter issue avoids any multivariable regression in the Saint Louis sample (indeed, it is commonly reported that at least 10 events should be observed when including one variable in the model). Then, to handle the first issue, lasso logistic regression - following its original use in linear models⁶³ - was used. It tends to produce some model coefficients that are exactly 0 and hence gives interpretable models. Thus, this technic is robust in this context of our analysis due to its tendency to prefer solutions with fewer non-zero coefficients, effectively reducing the number of features upon which the given solution is dependent”

References

Tibshirani, R. (1996). Regression shrinkage and selection via the lasso. J. Royal. Statist. Soc B., Vol. 58, No. 1, pages 267-288).

2- Page 2, Line 75: In epidemiology, controls are a study group who are likely to acquire the disease and have the same risk factors as the cases - which end up getting the disease. Calling the sibling donors "controls" is a confusing use of the term. The samples that the study team acquired from the transplant recipient who did not acquire GvHD at day 90 can be labeled as controls.

We thank the reviewer for this pertinent comment and we change the term “control” used for donors by the term “*healthy subjects*” to avoid any ambiguity. This point is close to reviewer 1’s point 2, as it highlights the fact that the metabolome described in each recipient is mainly produced by host (or host microbiota) metabolism rather than by immune donor cells circulating in recipients. This is an additional reason for removing the term “control” that we initially used for donor, that could lead to thinking that metabolites measured after transplantation are “donor-derived”.

3- Figure 2a: Using the word, "Filtration" is ambiguous. Please reword this sentence: Filtration of 83 metabolites ...: as "Discarded 83 metabolites ..."

This correction has been done for figure 2, figure 3, supplementary figure 1 and supplementary figure 2.

4- Page 2, Line 92: It is misleading to state that 37 and 45 metabolites were different between GvHD recipients and their donors in cohort 1 and 2. In Figure 2a, the statistical test with correction for false positives shows that only 5 were different between GvHD recipients and their donors in cohort 1. The corresponding number for cohort 2 is unclear because the tables in the supplementary tables are not correctly formatted. Please see suggestions for correcting this format below.

We now corrected the manuscript to mentioned only metabolites that were more frequently detected in one patient category after correction for multiple testing. We also modified table format to improve legibility.

“Using this approach, we identified 5 and 3 metabolites in cohort 1 and 2 that were more frequently detected in donors or in their related recipients after correction for multiple testing, respectively (supplementary tables 2 and 4).”

Presentation of Supplementary Tables

5- The supplementary tables are unreadable. Each table is split across two pages that are not sequential. Would the authors please reformat in landscape format on a larger area (i.e. A3 size or higher in PDF format) so that each row of a table is at least captured in a single page?

We apologize for this inconvenient. We submitted an excel file but it was converted in PDF during the submission process, that split the table on multiple pages. This has been corrected now.

6- Figure 3: Parts a + b shows that none of the metabolites are significantly different after correction for false positives after multiple hypothesis testing. In the text, Page 3, Line 122-125 state that a number of metabolites are "significantly different after Bonferroni correction". The figure does not show this finding. These results are thus inconsistent with the findings shown in the figure.

Sorry for this confusion, we reformulate the sentence to be more precise about this result. The initial sentence mentioned indole-propionate as the only metabolite that was significant after Bonferroni correction, but this result refers to cohort 2. It has been changed in the manuscript:

“Assessment of non-detectable value distribution among patients revealed that indole-propionate, a microbial-derived compound from tryptophan metabolism, was the only metabolite significantly less frequently detected at GvHD onset after Bonferroni correction in cohort 2 (figure 3a and supplementary figure 2a, supplementary table 6 and 8).”

7- Under Data Analysis: Page 6, Line 268-269: "The confirmatory analysis used the same filters and procedures than the exploratory analysis to ensure a true validation analysis." A true validation analysis would take the significantly different metabolites from the exploration stage and use these metabolites to predict disease state in an independent cohort. If the metabolites that were detected as different in the exploratory state were also found to be predictive of the disease state, then true validation of the metabolites association with disease could be claimed. In the absence of such an analysis, all that the authors have attempted to do is to test whether their findings were consistent across two independent cohorts. Recommend removing the phrase "true validation analysis". Alternatively, the two cohorts could have been combined in a single multivariable regression analysis with an indicator variable for cohort to test consistency and increase the statistical power of the analyses. Using multivariable analyses would be consistent with modern statistical approaches in the field.

We agree with this comment; our objective was not to identify biomarkers and to test them on a validation cohort. The second cohort was a confirmatory independent multicentric cohort of patient to confirm the consistency of findings described on the first cohort. To

avoid any misinterpretation of our study, we have removed the sentence “true validation analysis” from our manuscript and have modified the sentence as follows:

“The confirmatory analysis used the same filters and procedures than the exploratory analysis to assess whether the results obtained in the first cohort of patients were consistent in an independent cohort.”

Otherwise, we did not combine the two cohorts in a single multivariable analysis, since we do think that evaluating the robustness of such findings across independent samples is of prime interest. Otherwise, observed associations could be spurious or data-driven. This was exemplified in the common failure of replication of published genetic associations with behavioral traits (Benjamin et al., 2012; Hewitt, 2012).

8- Imputation Strategy: The method of imputation with a fixed value followed by addition of Gaussian noise introduces directional bias into the analyses. Have the authors considered a more robust form of imputation such as multiple imputation? Please see methods here: <https://www.ncbi.nlm.nih.gov/pmc/articles/PMC3074241/>

We completely agree with the importance of correct imputation methods of missing values to limit the biases in statistical analyses. Multiple imputation (MI) is indeed a well-established procedure to deal with missing data and has been established as one of the leading methodologies to deal with incomplete data. However, standard MI procedures for continuous missing values (described in the suggested reference above) are often unrestricted, meaning that imputed values are selected across the distribution of the observed data. Here, missing data rely on the limit of quantification (LOQ): any value below this limit cannot be measured and reported (BLQ data). Thus, missing data less than the LOQ should not be imputed from this observed range. The missingness process actually depends on missing values, suggesting missing not at random (MNAR) mechanism of missing data, avoiding the use of this multiple imputation approach. In other words, imputing BLQ data ignoring this information entails a loss of information and result in a bias toward higher values.

To consider the information of BLQ data – known as “left-censoring”-, several approaches have been proposed. One commonly used method is complete case analysis, where observations with values below the LOQ are simply eliminated. However, it is likely to be biased in other situations than data missing completely at random. Another easy method is single imputation, where every value below the LOQ is replaced by a constant such as $LOQ/2$ or $LOQ/\sqrt{2}$. It allows unbiased estimates unless a large amount of data is missing (but such cases were removed by the filtering, at the end of step 1 of the analysis). These findings were notably reported in Keizer et al. (Pharmacology Research & Perspectives, 2015) who showed that except when the percentage of missing data is high, imputing values by $LOQ/2$ achieves less bias than simply discarding the data, and in fact is similar to having complete data or using more complex methods than the ones used in this paper (see especially Fig. 3 on page 6). Its main drawback is to generate ties in the sample, thus resulting in impaired estimates of variances. Therefore, we used this single imputation but generating some

randomness. This method respects the important information that metabolite amounts are lower in these samples than in all other samples.

All these points have been more clearly stated in the revised manuscript by adding the following paragraph:

“Based on the experimental protocol, missing data can be considered all to rely on the limit of quantification (LOQ) of the analytical method: any value below this limit could not be measured and reported (BLQ data). Such missingness process actually depends on missing values, suggesting missing not at random (MNAR) mechanism of missing data. To consider the information of such BLQ data – known as “left-censoring”-, the most usual method is to impute half the LoQ to missing data. Since LoQ is unknown for each chemical, imputations used half the minimal observed value. It allows unbiased estimates unless a large amount of data is missing (but such cases were removed by the filtering, at the end of step 1 of the analysis). These findings were notably reported in Keizer et al.⁶¹ who showed that except when the percentage of missing data is high, imputing values by LOQ/2 achieves less bias than simply discarding the data, and in fact is similar to having complete data or using more complex methods than the ones used in this paper. Its main drawback is to generate ties in the sample, thus resulting in impaired estimates of variances. Therefore, a small random noise generated from a Gaussian distribution was added to imputed values, and then rounded to the nearest integer to maintain the characteristics of the original data. A 100 variance value of the Gaussian distribution was used to ensure a lower variability than that observed in detected amounts.”

Description of Study population

9- What are the clinical risk factors for GvHD outside of metabolites? Were these clinical risk factors equivalently represented in the transplant recipients with and without GvHD? Please present a table comparing the basic demographics of recipients in the two cohorts, the donors and a table comparing the demographics of recipients who acquired GvHD vs. recipients who did not acquire GvHD. Please include a comparison of clinical factors that were collected as part of the study (Page 5, Line 188 through Line 191)

Main GvHD factor risks are age, gender mismatch (female donor to male recipient) and HLA mismatch. We only included HLA-identical sibling donor/recipient pairs to avoid any biases due to HLA mismatch. Age, gender and gender mismatch, together with other main clinical variables are now described in supplementary table 1 for the GvHD and non GvHD groups in both cohorts, as requested by reviewers 2 and 3.

Minor Comments

10- Line 82: "The distribution of these metabolites ..." Which sets of metabolites are the authors referring to - the shared ones or across all metabolites? Please clarify. If shared, modify the sentence as: "The distribution of these shared metabolites ..."

We referred to shared metabolites, the sentence has been corrected.

“The distribution of these shared metabolites into each metabolic pathway was”

11- Lines 82-84: Do the percentages indicate average percentages or median percentages? Please clarify.

These percentages represent the average percentages of each super pathway in both cohorts. This has been now clarified in the manuscript. In addition, figure 1C was modified to improve legibility, as suggested by reviewer 3.

“The average distribution of these shared metabolites into each metabolic pathway was: lipid (46.18%), amino acid (24.16%), xenobiotics (13%), nucleotides (4.43%), carbohydrate (3.82%), cofactors and vitamins (3.82%), peptide (3.21%) and energy (1.38%) pathways (figure 1c).”

12- Page 2, Line 99: "fatty acid, mono and diacylglycerol and primary bile acid (BA)": do these correspond to the 5 metabolites that are shown as significantly different after correction for false positives after univariate multiple hypothesis testing?

The 5 metabolites that were significantly different after correction for false positive were identified by the analysis comparing the distribution of detected values among patient categories for each metabolite (figure 2A or supplementary figure 1A). The text line 99 refers to metabolites that were identified by comparing amounts of metabolites between patient categories (figure 2C-D of supplementary figure 1C-D). To avoid any misinterpretation, the text was corrected as followed:

“Metabolites with putative biological relevance were identified by comparison of the amount of each metabolite between recipients without GvHD and their paired related donors (figure 2c and d, supplementary figure 1c and d, and supplementary table 3 and 5). As compared with healthy subjects, allogeneic HSCT recipients without GvHD were mainly characterized by a significant increase in the amounts of complex lipid metabolites, especially fatty acid, mono and diacylglycerol and primary bile acid (BA).”

Reviewer #3:

This is a paper which has explored the metabolic changes of graft versus host disease (GvHD) after allogeneic stem cell transplantation both in comparison to the original donors and in comparison to a transplant group that did not experience GvHD. There are two independent cohorts examined in this study which should allow for more robust conclusions to be drawn from the results. In general, I believe the paper will be of interest to the medical community, and its potential insights into the effects of graft versus host disease may be of interest to the wider immunological research community.

I have some minor concerns about the data analysis that has been conducted and some major concerns about the reporting of the metabolomics experiments.

General comments

1- Figures: Much of the figure text is too small to read comfortably and should be increased to size 12 Arial or equivalent.

Thanks for this comment; we increased police size as much as possible to improve legibility of figures.

2- Tables: All of the tables featuring metabolite data fail to report important information on the average detected intensity for each metabolite per class and the range per class. Standard deviation should also be reported.

As requested, we corrected all supplementary tables to provide information about mean intensity and standard deviation of significant metabolites. This was calculated on raw observed intensity, before any missing value imputation, to give an objective view of the real amount of each metabolite that were selected by statistical test. It is important to note that statistical test (p value and fold change value in the table), as explained in methods, were done after missing value imputation. As a consequence, the fold change can be different from what would be calculated with the ratio of mean values that are now reported in the table. As suggested in of reviewer 3's last comment, we also removed metabolites with high fold change but who were not statistically significant.

3- Statistics: You have compared the D v R response and the Ra v. Rs response but nowhere have you reported the results of the D v R comparison for the two classes of recipient. I think the latter is an important comparison given the general variability in human data – the use of sibling pairs will remove some of this effect and may prove insightful. I would have anticipated some assessment of confounding factors (age and BMI being the two major possibilities) with some statistical approach used to account for this. An attempt to do this retrospectively by pairing samples was done but no details were given on how well matched these pairs were. BMI could play an important role in some of the results, especially as it is known to be correlated with both lipid levels and the fecal microbiome.

In the first version of the manuscript, we focused our analysis on the comparison of donors and Rs, to described metabolomics variations associated with transplantation in the absence of GvHD. But we agree with this comment and we now added the comparison of Ra patients (with GvHD) with their related donors, to allow the description of the similar comparison in case of GvHD. This has now been added to the result section and we added two figures (figure 3 and supplementary figure 2, and supplementary table 6, 7 and 8)

The following paragraph was added to results:

“Using the same approach, we compared metabolomic profiling of recipients with GvHD to their related donors. Few metabolites were more frequently detected in donors or in recipients with GvHD ($n=0$ in cohort 1 and $n=6$ in cohort 2) (figure 3a and supplementary figure 2a, supplementary table 6). Comparison of the amounts of each metabolite identified 150 and 182 metabolites that were significantly changed after transplantation, in cohort 1 and 2 respectively, with 110 metabolites shared by both cohorts (figure 3c-d and supplementary figure 2c-d, supplementary table 7 and 8). Most of metabolic pathways involved in recipients with GvHD were similar to those identified in recipients without GvHD, with the exception of primary BA and mono/diacylglycerol that were increased in recipients by comparison with their donors. A common feature of all recipients was a strong decrease in xenobiotics detection, especially in xanthine, tobacco and food metabolites. This suggests that all recipients modified their behaviors regarding food or tobacco intake, as strongly recommended after HSC transplantation.”

To consider potential confounders, that is, age, sex and BMI, analyses were performed both unadjusted or adjusted for BMI (see changes in the « statistical methods » part).

“To handle obvious confounders, all analyses of steps two and three (detailed below) were secondly adjusted. More specifically, unpaired analyses were adjusted for age, sex, and BMI. Analyses on paired cases (either D vs Rs, or D vs Ra), were adjusted for differences of age and of BMI, as well as for sex match between the two subjects of the pair. For paired analysis of Ra vs paired Rs, given matching was performed on age and sex, estimates were only adjusted for difference in BMI between the two subjects of a pair.”

Despite some metabolite changes between groups were no longer significant after this adjustment, most of the findings regarding the metabolites discussed in the text were not markedly modified.

4- You have assessed absolute differences in missing values between the sample classes but have not assessed whether this missingness could in fact be related to the average detected intensity of the metabolite in question. If these are poorly detected metabolites, you may be assessing only that they are barely reaching the limit of detection.

Equally, you could be measuring poorly controlled storage and collection conditions – as the protocols for these were not given, it cannot be judged.

To avoid any doubt regarding sample storage and collection conditions, we now added this information to methods, as suggested in the comments below. We also did not detect outlier that could reflect technical problems with some samples. The fact that some metabolites are sometimes not detected in some sample does not seem to be linked to the average intensity of the corresponding metabolites (see for your information the graph below, correlation coefficient = -0.101 and -0.053 respectively, for cohort 1 and 2).

Cohort 1:

Cohort 2:

5- You have removed data for which 50% of values are missing, but have used half minimum values to substitute the remainder of missing values (MV). This will potentially severely bias your results, especially when there appears to have been no assessment on whether the missing values were correlated with intensity or not. I would strongly recommend consulting an independent statistician for guidance on this. Even if the approach was justified, imputation may not be required for univariate statistical methods and it will bias your results less to type 1 errors if it is avoided.

Missing data on metabolites were handled through two main approaches.

First, as reported by the reviewer, we discarded all metabolites that were missing at least in 50% of individuals in either group from Saint Louis cohort. This could not be responsible for any bias, given no patient were removed but only metabolites. However, we agree with the Reviewer that, doing so, we could have avoided the possibility of describing those metabolites as potentially predicting GvHD. Nevertheless, not to ignore those metabolites, they were all considered in the analysis of “not detected” versus “detected” metabolites in the reported analyses.

Secondly, based on the experimental protocol, missing data can be considered all to rely on the limit of quantification (LOQ) of the analytical method: any value below this limit could not be measured and reported (BLQ data). Such missingness process actually depends on missing values, suggesting missing not at random (MNAR) mechanism of missing data. To consider the information of such BLQ data – known as “left-censoring”-, several approaches have been proposed. One commonly used method is complete case analysis, where observations with values below the LOQ are simply eliminated. However, it is likely be biased in other situations than data missing completely at random such as in our setting. We therefore applied single imputation, where every value below the LOQ is replaced by a $LOQ/2$. It allows unbiased estimates unless a large amount of data is missing (but such cases were removed by the filtering, at the end of step 1 of the analysis). These findings were notably reported in Keizer et al. (Pharmacology Research & Perspectives, 2015) who showed that except when the percentage of missing data is high, imputing values by $LOQ/2$ achieves

less bias than simply discarding the data, and in fact is similar to having complete data or using more complex methods than the ones used in this paper (see especially Fig. 3 on page 6). Its main drawback is to generate ties in the sample, thus resulting in impaired estimates of variances. Therefore, we used this single imputation but generating some randomness.

All these points have been more clearly stated in the revised manuscript by adding the following paragraph:

“Based on the experimental protocol, missing data can be considered all to rely on the limit of quantification (LOQ) of the analytical method: any value below this limit could not be measured and reported (BLQ data). Such missingness process actually depends on missing values, suggesting missing not at random (MNAR) mechanism of missing data. To consider the information of such BLQ data – known as “left-censoring”-, the most usual method is to impute half the LoQ to missing data. Since LoQ is unknown for each chemical, imputations used half the minimal observed value. It allows unbiased estimates unless a large amount of data is missing (but such cases were removed by the filtering, at the end of step 1 of the analysis). These findings were notably reported in Keizer et al.⁶¹ who showed that except when the percentage of missing data is high, imputing values by LOQ/2 achieves less bias than simply discarding the data, and in fact is similar to having complete data or using more complex methods than the ones used in this paper. Its main drawback is to generate ties in the sample, thus resulting in impaired estimates of variances. Therefore, a small random noise generated from a Gaussian distribution was added to imputed values, and then rounded to the nearest integer to maintain the characteristics of the original data. A 100 variance value of the Gaussian distribution was used to ensure a lower variability than that observed in detected amounts.”

6- Methods; You have not followed good reporting practice for metabolomics methods as laid out in Sumner, L.W., Amberg, A., Barrett, D. et al. Metabolomics (2007) 3: 211. <https://doi.org/10.1007/s11306-007-0082-2> and Goodacre, R., Broadhurst, D., Smilde, A.K. et al. Metabolomics (2007) 3: 231. <https://doi.org/10.1007/s11306-007-0081-3>. The details provided in your methods are not suitable for someone else to be able to replicate your experiment to any degree and thus fall short of the requirements of good scientific reporting.

This is perfectly true and we apologize for the lack of details provided in the first version of the manuscript. As requested by reviewer 3, we added additional informations regarding technical aspect of the procedure that are now added to methods and discussed in the following specific points raised by the reviewer: see responses below in the different paragraphs; we responded to every queries to fit with the good reporting practice according to the suggested references.

- First of all, we corrected the “mass spectrometry” paragraph in the “Methods” section as follow:

“All methods utilized a Waters ACQUITY ultra-performance liquid chromatography (UPLC) and a Thermo Scientific Q-Exactive high resolution/accurate mass spectrometer interfaced with a heated electrospray ionization (HESI-II) source and Orbitrap mass analyzer operated at R = 35,000 mass resolution. The sample extract was dried then reconstituted in solvents

compatible to each of the four methods. Each reconstitution solvent contained a series of standards at fixed concentrations to ensure injection and chromatographic consistency. One aliquot was analyzed using acidic positive ion conditions, chromatographically optimized for more hydrophilic compounds. In this method, the extract was gradient eluted from a *C18-column (Waters UPLC BEH C18-2.1x100 mm, 1.7 μm)* using water and methanol, containing 0.05 % perfluoropentanoic acid (PFPA) and 0.1% formic acid (FA). A second aliquot was also analyzed using acidic positive ion conditions, however it was chromatographically optimized for more hydrophobic compounds. In this method, the extract was gradient eluted from the same afore mentioned C18 column using methanol, acetonitrile, water, 0.05 % PFPA and 0.01 % FA and was operated at an overall higher organic content. A third aliquot was analyzed using basic negative ion optimized conditions with a separate dedicated C18 column. The basic extracts were gradient eluted from the column using methanol and water, however with 6.5mM ammonium bicarbonate at pH 8. The fourth aliquot was analyzed via negative ionization following elution from a *HILIC column (Waters UPLC BEH Amide 2.1x150 mm, 1.7 μm)* using a gradient consisting of water and acetonitrile with 10 mM ammonium formate, pH 10.8. The MS analysis alternated between MS and data-dependent MSⁿ scans using dynamic exclusion. The scan range varied slightly between methods but covered 70-1000 *m/z*.”

Specific comments:

7- Table 1: There is a major confounding factor of the average day at which blood was drawn from Ra (<23 or 28 days) vs Rs (90 days) – this was not highlighted or accounted for in the statistics or discussion.

This is a very important point also raised by reviewers 1 which has now been added to discussion. For feasibility reasons, we had to define a timepoint for patients without GvHD that was set as day + 90, after transplantation. Patients with GvHD were sampled at GvHD onset, before corticosteroid treatments, resulting in slightly different delays between transplantation and sampling (mean delay day+40 in cohort 1 and day+33 in cohort 2, median delay day+26 in cohort 1 and day+28 in cohort 2). Even if we cannot exclude that this difference could contribute to variations observed between patients with or without GvHD, the impact of difference could be minored by the fact that almost all patient had natural oral feeding, similar antibiotics treatments and received the same immunosuppressive drugs at the time of sample. For immunosuppression, all samples were obtained before corticosteroid treatment and all patients (with or without GvHD) received GvHD prophylaxis based on calcineurin inhibitors at the time of sample (that was cyclosporine in more than 90% of patients).

The following sentences were added to the discussion:

“Graft versus host disease is an immune reaction from donor immune cells targeting allogeneic antigens in recipients, whose pathophysiology is still only partially understood in Humans. Recent experimental researches have highlighted the complex network of interactions and regulations between immune cells, microbiota and host environment⁴. Here we report that patients who underwent allogeneic HSCT have major metabolomics changes compared to healthy subjects, and that acute GvHD onset seems to be associated with specific differences in metabolomics profile by comparison with patients who did not developed GvHD. Comparing patients without GvHD with those who developed an unpredictable time onset GvHD, leads to intrinsic differences between groups of patients that

may have contributed to the observed variations in metabolites. These differences include: slight differences in the day of sampling after transplantation, feeding or treatments received at the time of sampling. To minimize the putative impact of these confounding factors, we used two independent cohorts of patients to confirm our results and explored groups of as much as possible similar patient characteristics in terms of HLA-matching, immunosuppression or diet mode.”

8- There are no statistics on the differences between the donors and recipients for BMI. There are no statistics on the difference of Ra vs Rs groups given for either cohort.

These points had now been added to supplementary table 1. Median BMI were 24.4 and 22.3 kg/m² (p=.07) for Rs and Ra patients respectively in cohort 1 and 25.6 and 25.7 (p=.36) respectively in cohort 2. For donors of patients Rs and Ra, median BMI were 26.2 and 22.2kg/m² (p=.1) respectively in cohort 1 and were not available in cohort 2. Additional relevant variables for Rs and Ra patients are now compared in the same supplementary table 1.

9- Given the importance attached to the tryptophan pathway, there is no discussion on potentially different diets between groups, especially dietary fiber intake. This may not have been collected at the time, but I would have anticipated a discussion on it as a confounding factor in the results.

We did not have detailed information about dietary fiber intake or detailed food intake. However, we know that most patients received oral feeding (95% of recipients). Metabolites detected in recipients by comparison with their related donors suggested some difference in eating behaviors, mainly in xanthine metabolism (suggesting different consumption of chocolate, tea and coffee) or in condiments (piperine, alliin, cinnamoilglycine). However, we did not observe major variations in betainized compounds that could have reflect a higher consumption of whole-grain rather than refined grain. This is now discussed in the manuscript.

“Our results suggest that transplantation is followed by modification in the recipients’ eating behaviors, mainly affecting xanthine metabolism (suggesting different consumption of chocolate, tea and coffee) or condiments-derived metabolites (piperine, alliin, cinnamoilglycine). Importantly, most patients had oral feeding at time of sampling and did not receive artificial feeding that could have affected metabolism.”

10- Line 80 There is a mismatch between the figures reported here and the figures reported in Fig 1b. Why?

We apologize for this error and we corrected the figure and the main text. The true number of metabolites detected in cohort 1 and 2 were 801 and 927 respectively, with 653 metabolites shared by both cohorts.

11- Figure 1c: Illegible

This figure has now been changed to improve legibility. Sub-pathways’ names were replaced by numbers; correspondence between numbers and pathways is available in the legend for figure.

12- Supp figures 3 and 5 describe themselves as cohorts 1 and 2 from St Louis. Is this a mistake or were there two cohorts from St Louis?

This is an error in the title of supplementary Table 5, cohort 2 is the multicentric cohort. It has been now corrected; sorry.

13- Line 104 „BA, tryptophan and polyamine metabolites were the pathways most affected (Fig 2 and supp Fig 1). The polyamines do not seem from the figures given here to be unduly affected and there is only one metabolite from the tryptophan pathway listed. The supplementary tables give more evidence to this statement.

The network analysis in figure 2e-f and supplementary figure 1e-f graphically represents the ratios between each metabolite that are directly connected, if this ratio is not significantly different between groups of patients. It confirms the univariate analysis showing that BA and tryptophan pathways are the most affected after transplantation as they appear at each extremity of the main graph. The fact that only one metabolite from tryptophan pathway appears disconnected from the whole graph suggests that indolepropionate variation is independent from all other metabolites, but does not exclude that amounts of other metabolites from the tryptophan pathway are also significantly different between donors and recipients. If polyamines are identified as significantly different between donors and recipients in both cohorts, we agree that the network analysis does not highlight as well the independence of polyamine. For this reason, we corrected the sentence line 103 as follow:

“Global analysis of metabolites behavior by comparison of their relative amount confirmed that BA and indole propionate were the most affected after transplantation”

14- Figure 4a: text too small and no error bars provided on radar plots.

We have now corrected figure 4A by increasing text size and adding error bar for each metabolite, in both cohorts.

15- Figure 4b: the ligands are not differentiated. Indole propionate is also not a dose responsive ligand of AHRs (e.g. see Hubbard et al 2015).

It has now been added to the figure to better identify metabolites on each plot. We also thank the reviewer for this interesting reference that was now added to the paper and to the discussion. It is clear that the difference that we report here in the amount of each AhR ligands does not reflect directly the biological consequence for the recipients. We agree that considering dose-responsive effect of each ligand is a really important point for discussion. That is also why we think that the comparison of detected versus undetected metabolites (figure 3A and supplementary figure 2A) is an unusual but interesting analysis that can highlight the fact that some metabolites are not only lower in patients with GvHD but also more frequently undetectable in these patients.

The following sentences have been added to discussion as well as two references from Hubbard et al. (Sci. Report, 2015 and Drug Metab. Dispos Biol. Fate Chem., 2015):

“However, it should be emphasized that the functional consequence of AhR ligands decrease should be explored, best in animal models. In fact, it has been demonstrated experimentally that the biological effect of some AhR ligands, such as indole-propionate, is not dose-responsive^{40,41}, suggesting that the absence of some AhR ligands could be more relevant than their decrease in acute GvHD. This is also why it appears that the comparison of metabolites that are detected or not detected in the recipients (figure 3A, supplementary figure 2A, table 6 and 8) is, in our view, as important as the comparison of metabolites amount. Using this approach, we identify N-acetyl-kynurenine in both cohorts, as well as indole-3-carboxylic

acid, 3-indoxyl sulfate and indole-propionate in cohort 2, that were not only decrease in recipients with GvHD but also more frequently undetectable in these patients.”

16- Line 204: protocol fails to give anticoagulant, time or temperature of processing

This point has now been added to methods section. *All samples were collected on EDTA tubes (BD Vacutainer, K3E 7.2mg, Plus blood Collection Tubes), centrifuged within 4 hours to collect plasma and immediately frozen at -80°C until processing.*

17- Line 209: how much plasma was eventually extracted and what were the recovery standards used and at what concentrations?

This point has now been added to methods section. *“Aliquots of 1 mL were divided in four aliquots of 250 µL for further study and sent to Metabolon company (Morrisville, US) for further process, using standardized processes as previously described⁵⁸. Samples were prepared using the automated MicroLab STAR® system from Hamilton Company. Several recovery standards were added prior to the first step in the extraction process for QC purposes. For the metabolomic analysis, a total of 100 microliters of sample was extracted under vigorous shaking for 2 min (Glen Mills GenoGrinder 2000) with methanol containing the following recovery standards: DL-2-fluorophenylglycine, tridecanoic acid, d6-cholesterol, and DL-4-chlorophenylalanine. The resulting extract was divided into five fractions: two for analysis by two separate reverse phase (RP)/UPLC-MS/MS methods with positive ion mode electrospray ionization (ESI), one for analysis by RP/UPLC-MS/MS with negative ion mode ESI, and one for analysis by HILIC/UPLC-MS/MS with negative ion mode ESI. The remaining aliquot was reserved for backup. Samples were placed briefly on a TurboVap® (Zymark) to remove the organic solvent. The sample extracts were stored overnight under nitrogen before preparation for analysis.”*

18- Line 222 which solvents were the samples reconstituted with?

This point has now been added to methods section. Four aliquots of each sample were taken from the extract and dried. Aliquots destined for each of the analyses were reconstituted in solutions designed for that specific chromatographic system. The solutions also contained a series of internal standards used to monitor instrument performance and allow alignment of the resulting data. We added more details to methods, “Mass spectrometry” paragraph: *“For each sample, two aliquots of each sample were reconstituted in 50 µL of 6.5 mM ammonium bicarbonate in water (pH 8) for the negative ion analysis and another two aliquots of each were reconstituted using 50 µL 0.1% formic acid in water (pH ~3.5) for the positive ion method.”*

19- Line 223 which standards were used and at what concentrations?

This point has now been added to methods section: *“The internal standards consist of a variety of deuterium labeled or halogenated biochemicals specifically designed both to cover the entire chromatographic run and to not interfere with the detection of any endogenous biochemicals. Authentic standards of d7-glucose, d3-leucine, d8-phenylalanine and d5-tryptophan were purchased from Cambridge Isotope Laboratories (Andover, MA). D5-hippuric acid, d5-indole acetic acid and d9-progesterone were procured from C/D/N Isotopes, Inc. (Pointe-Claire, Quebec). Bromophenylalanine was provided by Sigma-Aldrich Co. LLC. (St. Louis, MO) and amitriptyline was from MP Biomedicals, LLC. (Aurora, OH). Recovery standards of DL-2-fluorophenylglycine and DL-4-chlorophenylalanine were from Aldrich*

Chemical Co. (Milwaukee, WI). Tridecanoic acid was purchased from Sigma-Aldrich (St. Louis, MO) and d6-cholesterol was from Cambridge Isotope Laboratories (Andover, MA). Standards for the HILIC dilution series of alpha-ketoglutarate, ATP, malic acid, NADH and oxaloacetic acid were purchased from Sigma-Aldrich Co. LLC. (St. Louis, MO) while succinic acid, pyruvic acid and NAD⁺ were purchased from MP Biomedicals, LLC. (Santa Ana, CA)".

Limit of Detection (LOD) for Standards analyzed in a Dilution Series Using Reverse-Phase Chromatography:

Standard	HRAM LOD ng/mL	UMR LOD ng/mL
d7-glucose	1.0	50.0
d3-leucine	0.25	5.0
d8-phenylalanine	0.25	3.0
d5-tryptophan	0.25	25.0
d5-hippuric acid	0.25	5.0
Br-phenylalanine	0.25	3.0
d5-indole acetic acid	3.0	25.0
amitriptyline	0.5	3.0
d9-progesterone	1.0	25.0

”

20- Line 218 to 236 please give exact details of gradients and conditions used as is concordant with standard reporting practices for UPLC-MS. Line 235 please give details of data dependent conditions

While the details of the chromatographic gradient were not included in the methods, information about Metabolon’s chromatography methods have been published in a paper referenced in the original submission (Evans AM, Bridgewater BR, Liu Q, Mitchell MW, Robinson RJ, et al. (2014) High Resolution Mass Spectrometry Improves Data Quantity and Quality as Compared to Unit Mass Resolution Mass Spectrometry in High-Throughput Profiling Metabolomics. *Metabolomics* 4:132. doi:10.4172/2153-0769.1000132. We added this reference to the paper and gave more details in the “mass spectrometry” paragraph of the “methods” section:

“One aliquot was analyzed using acidic positive ion conditions, chromatographically optimized for more hydrophilic compounds. In this method, the extract was gradient eluted from a C18 column (Waters UPLC BEH C18-2.1x100 mm, 1.7 μm) using water and methanol, containing 0.05 % perfluoropentanoic acid (PFP) and 0.1% formic acid (FA). A second aliquot was also analyzed using acidic positive ion conditions, however it was chromatographically optimized for more hydrophobic compounds. In this method, the extract was gradient eluted from the same afore mentioned C18 column using methanol, acetonitrile, water, 0.05 % PFP and 0.01 % FA and was operated at an overall higher organic content. A third aliquot was analyzed using basic negative ion optimized conditions with a separate dedicated C18 column. The basic extracts were gradient eluted from the column using methanol and water, however with 6.5mM ammonium bicarbonate at pH 8. The sample injection volume was 5 μL

and a 2x needle loop overfill was used. Separations utilized separate acid and base-dedicated 2.1 mm x 100 mm Waters BEH C18 1.7 μ m columns held at 40°C. The fourth aliquot was analyzed via negative ionization following elution from a HILIC column (Waters UPLC BEH Amide 2.1x150 mm, 1.7 μ m, held at 40°C) using a gradient consisting of water (15%), methanol (5%) and acetonitrile (80%) with 10 mM ammonium formate, pH 10.8.”

21- Line 238: which ions were used for identification and what quality controls were conducted to ensure they were accurate, especially with regard to retention time matching to a library. How many ions were used as the minimum qualifying ions for identification

Spectra were collected using alternating MS and data dependent MS/MS scans. The MS/MS scans were collected based on the abundance of ions in the preceding spectra and included the use of both a static exclusion list of known background ions and a dynamic exclusion list to prevent over representation of the most abundant biochemicals.

Metabolite identifications were made based on comparison of MS spectra, chromatographic retention, and fragmentation to a comprehensive in-house library built from authentic standards. The library entries were built using the exact methods used in the current study to assure the greatest comparability between the experimental values and the library values. Additionally, every metabolite assignment was manually approved, and the assignment was confirmed by a second reviewer.

The following corrections were done in the “compounds identification” paragraph, methods sections:

“Compounds identification and quantification

Raw data was extracted, peak-identified and QC processed using Metabolon’s hardware and software. Compounds were identified by comparison to library entries of purified standards or recurrent unknown entities, as previously described^{57,58}. Briefly, Metabolon maintains a library based on authenticated standards that contains the retention time/index (RI), mass to charge ratio (m/z), and chromatographic data (including MS/MS spectral data) on all molecules present in the library. Furthermore, biochemical identifications are based on three criteria: retention index within a narrow RI window of the proposed identification, accurate mass match to the library +/- 10 ppm, and the MS/MS forward and reverse scores between the experimental data and authentic standards. The MS/MS scores are based on a comparison of the ions present in the experimental spectrum to the ions present in the library spectrum. While there may be similarities between these molecules based on one of these factors, the use of all three data points can be utilized to distinguish and differentiate biochemicals. More than 3300 commercially available purified standard compounds have been acquired and registered for analysis on all platforms for determination of their analytical characteristics. Microbiota-derived metabolites identification was based on the Human Metabolome Database (www.hmdb.ca). The QC and curation processes were designed to ensure accurate and consistent identification of true chemical entities, and to remove those representing system artifacts, mis-assignments, and background noise. Metabolon data analysts use proprietary visualization and interpretation software to confirm the consistency of peak identification among the various samples. Library matches for each compound were checked for each sample and corrected if necessary. Peaks were quantified using area-under-the-curve. A data normalization step was performed to correct variation resulting from instrument inter-day tuning differences. Essentially, each compound was

corrected in run-day blocks by registering the medians to equal one (1.00) and normalizing each data point proportionately.”

22- Line 250: please give details of what quality controls and quality assurance you used to ensure the mass spectrometry was robust. How many quality control samples and what was the median standard deviation of these quality control samples throughout the run?

How was the run randomised?

Three types of controls were analyzed in concert with the experimental samples: samples generated from a small portion of each experimental sample served as technical replicate throughout the data set; extracted water samples served as process blanks; and a cocktail of standards spiked into every analyzed sample allowed instrument performance monitoring. Instrument variability was determined by calculating the median relative standard deviation (RSD) for the standards that were added to each sample prior to injection into the mass spectrometers (median RSD = 3-4%). Overall process variability was determined by calculating the median RSD for all endogenous metabolites (i.e., non-instrument standards) present in 100% of the MTRX samples, which are technical replicates created from a large pool of extensively characterized human plasma. The median RSD for the MTRX samples was equal to 9-10%. Five MTRX samples and three process blank samples were processed per every batch of 30 samples. Experimental samples were randomized across the platform run with QC samples spaced evenly among the injections.

The following paragraph was added to the “methods” section:

Quality assurance and quality control (QA/QC)

Several types of controls were analyzed in concert with the experimental samples: a pooled matrix sample generated by taking a small volume of each experimental sample (or alternatively, use of a pool of well-characterized human plasma) served as a technical replicate throughout the data set; extracted water samples served as process blanks; and a cocktail of QC standards that were carefully chosen not to interfere with the measurement of endogenous compounds were spiked into every analyzed sample, allowed instrument performance monitoring and aided chromatographic alignment.

Instrument variability was determined by calculating the median relative standard deviation (RSD) for the internal standards that were added to each sample prior to injection into the mass spectrometers (median RSD = 3-4%). Instruments are calibrated at least weekly in the utilized polarity using Thermo and mass accuracy is monitored at the batch level for the internal standards. A batch fails QC if any of the internal standards are more than 5ppm away from the theoretical mass. Overall process variability was determined by calculating the median RSD for all endogenous metabolites (i.e., non-instrument standards) present in 100% of the MTRX samples, which are technical replicates created from a large pool of extensively characterized human plasma. The median RSD for the MTRX samples was equal to 9-10%. Five MTRX samples and three process blank samples were processed per every batch of 30 samples. Experimental samples were randomized across the platform run with QC samples spaced evenly among the injections.

23- What was the exact data processing method used – normalization, peak picking, noise reduction, filtering, software used?

In house peak detection and integration software was used whose data output was a list of m/z ratios, retention indices and area under the curve (AUC) values. The process is outlined

in the following publications that were added to references and additional description of compounds identification and quantifications methods were provided in the manuscript, as mentioned above:

*DeHaven CD, Evans AM, Dai H, Lawton KA (2010) Organization of GC/MS and LC/MS metabolomics data into chemical libraries. J Cheminform 2:9.*

*DeHaven CD, Evans AM, Dai H, Lawton KA (2012) Software Techniques for Enabling High-Throughput Analysis of Metabolomics Datasets. InTech Open.*

Normalization was performed as described above. A data normalization step was performed to correct variation resulting from instrument inter-day tuning differences. Essentially, each compound was corrected in run-day blocks by registering the medians to equal one (1.00) and normalizing each data point proportionately.

24- Were metabolites exactly quantified or only relatively quantified. To which standards?

Metabolites were relatively quantified based on library entries created from authenticated standards ran on each of the mass spectrometry methods.

25- Line 263: you have excluded both drugs and nutrients (is tryptophan not a nutrient???) but both are important to the eventual findings and should have been tested as possibly influencing factors on the results.

We corrected methods to avoid misinterpretation about metabolites exclusion for this analysis. Drugs were excluded but not nutrient. We decided to discard drugs as they were more the consequence in the difference of clinical status rather than relevant biological difference between groups of patients.

The following sentence is now mentioned in statistics methods:

“Only “natural” metabolites, including Bacteria and Fungi-derived metabolites, were considered, excluding drugs or assimilated compounds.”

26- Line 276-288: this seems a very simplistic approach to me. Results should have been at least correlated with intensities, or weighted in the scoring based on the average intensity.

This first step of analysis was done to avoid any biases that could be induced by the classical statistic approach, based on a pre-processing step of filtration and missing-value imputation step for compounds amount comparison. We considered that the distribution of detectable compounds between groups of patients, irrespective of the intensity, could bring a first level of information and would detect metabolites that are more frequently or only detected in one group of patients. As discussed in the comment above about the dose-responsive effect of indole-propionate, it can highlight the fact that irrespective of the amount of each metabolite, their presence or absence in patients can also be important to consider. All these metabolites were then more classically reanalyzed by comparison of the average amounts for each compound.

27- Line 293: see earlier comment about MV imputation

We completely agree with the importance of a correct imputation method of missing values in order to limit the biases in statistical analyses. Multiple imputation (MI) is indeed a well-established procedure to deal with missing data and has been established as one of the leading methodologies to deal with incomplete data. However, standard MI procedures for continuous missing values (described in the suggested reference above) are often

unrestricted, meaning that imputed values are selected across the distribution of the observed data. Here, missing data rely on the limit of quantification (LOQ): any value below this limit cannot be measured and reported (BLQ data). Thus, missing data less than the LOQ should not be imputed from this observed range. The missingness process actually depends on missing values, suggesting missing not at random (MNAR) mechanism of missing data, avoiding the use of this multiple imputation approach. In other words, imputing BLQ data ignoring this information entails a loss of information and result in a bias toward higher values.

To consider the information of BLQ data – known as “left-censoring”-, several approaches have been proposed. One commonly used method is complete case analysis, where observations with values below the LOQ are simply eliminated. However, it is likely be biased in other situations than data missing completely at random. Another easy method is single imputation, where every value below the LOQ is replaced by a constant such as $LOQ/2$ or $LOQ/\sqrt{2}$. It allows unbiased estimates unless a large amount of data is missing (but such cases were removed by the filtering, at the end of step 1 of the analysis). These findings were notably reported in Keizer et al. (Pharmacology Research & Perspectives, 2015) who showed that except when the percentage of missing data is high, imputing values by $LOQ/2$ achieves less bias than simply discarding the data, and in fact is similar to having complete data or using more complex methods than the ones used in this paper (see especially Fig. 3 on page 6).

Its main drawback is to generate ties in the sample, thus resulting in impaired estimates of variances. Therefore, we used this single imputation but generating some randomness. This method respects the important information that metabolite amounts are lower in these samples than in all other samples.

All these points have been more clearly stated in the revised manuscript.

28- Line 312: metabolites with a high variance may have large mean fold changes that are meaningless when they are not robust in their measurement e.g. high technical or natural biological variance. Identifying high fold changes alone may just be adding noise to resulting interpretation.

We agree with this comment and we now removed these compounds from tables and volcano plot. We only kept metabolites that were statistically significant. The sentence “Metabolites that showed an important fold change (more than doubling their amount, considering the geometric mean) were also identified, even when not significant.” was removed from the manuscript.

Reviewers' comments:

Reviewer #1 (Remarks to the Author):

Appreciate the good faith effort in answering the critiques adequately.

Reviewer #3 (Remarks to the Author):

The amendments that the authors have made to the first version strengthen the paper. I have only a few minor comments to make:

Line 170: Thank you for adding the point about dietary changes which add an additional perspective. I was in fact referring to the potential for dietary differences between the Ra and the Rs groups leading to the observed differences – this may have a causal factor, but may equally be a confounding factor in data interpretation.

Line 266 : what was the ratio or amount of methanol used? What was the concentration of recovery standards used.

Line 282: what were the concentrations of internal standards used? Were they used only for retention time indexing or also to normalize data?

Line 298: The paper you mention you have added to the manuscript does not seem to be either in the reference list or referred to in the methods section. The identically titled Bridgewater 2014 paper is listed in the references but not referenced in the text. I could find no specific reference to a previously published method. Please add the reference if wished (see note below) and include the percentage of each solvent and the solvent gradients used.

NB. The reference you gave has also been wrongly cited as

Evans AM, Bridgewater BR, Liu Q, Mitchell MW, Robinson RJ, et al. (2014) High Resolution Mass Spectrometry Improves Data Quantity and Quality as Compared to Unit Mass Resolution Mass Spectrometry in High-Throughput Profiling Metabolomics. *Metabolomics* 4:132. doi:10.4172/2153-0769.1000132.

This gives the impression it has been published in the highly regarded “Metabolomics” journal published by the noted academic publishing group Springer. The only paper I could find that matches the reference above was published in “Metabolomics: Open Access”, an Omics International journal of unknown reputation. The Bridgewater 2014 paper has also been published by this group. I believe Omics International is regarded by many as a predatory publisher (e.g., see Masic et al(1)). While this

does not exclude that they may be excellent papers scientifically, there may be a question mark about the amount and quality of any peer review that they underwent.

If you choose to use this reference, please update the journal name to reflect the correct journal.

Supp Table 1. I would recommend defining NA as meaning not available as opposed to not applicable to avoid confusion.

Line 318: no reference or details for this cocktail of QC standards is given. Please also define the abbreviation MTRX.

Line 334 onwards. The identifying list of MSMS is a key point to being able to reproduce the experimental conditions. Without them being listed, no reader has the opportunity of being able to review any possible problems with identification, especially of key metabolites. Ideally the full list of identifying features should be published in supplementary, at least for the metabolites listed as being of key importance. Alternatively you could reference another peer-reviewed article where this has already been done (the references provided do not give enough details). I understand the commercial problems this request may cause for Metabolon. This must be an editorial decision as to whether the current methods included here is an acceptable compromise.

Line 366: I presume that the data handling for D v Rs is identical as to that for D v Ra. Please update the text to reflect.

Line 411: Lastly, there are no easy answers to the perennial problem of missing values. I agree entirely with you that with 50% missing values, leaving out data would be equally problematic; we tend to have a much higher initial filter when we adopt this method. Thank you for bringing the interesting paper by Keizer et al to my attention. The data concerns pharmacokinetic data, rather than metabolomics, and the highest number of samples they had below the LOQ was 40% which they had determined was a „very high“ number of missingness. I notice that they conclude that likelihood-based imputation performs better than LLOQ/2. This is a similar conclusion to other papers in the field e.g. Wei et al 2019(2) and Do et al 2018(3).

Do et al discuss the issue of MNAR values as being the most likely for their data which, in the context of pharmacokinetics, makes sense. It is also seen in many metabolomics datasets for logical reasons. I have also dealt with data where ion suppression effects meant that the majority of missing data was missing at random (MAR). With your specific data, the correlation plots of MV versus intensity that you printed in your response were difficult to fully interpret as the y axes scale was a little stretched. However, there appeared to be no correlation between intensity and missingness by your own statistical analysis as answer to point 4 in the previous review (correlation coefficients of -0.1 and -0.05). Allowing for the possibility that some metabolites may be highly intense in some individuals and almost non-existent in others, the results would still point to missing values having a high proportion of MAR values – hence suggesting that other approaches such as random forest or k-nearest neighbour imputation would be preferable to half minimum value. Perhaps the compromise situation is to draw the readers attention to the potential bias of the approach used and highlight the percentage of missing values, the average detected intensity (excluding missing values) and the limit of detection for each class for metabolites found to be of major significance. I

would recommend that the new paragraph you added at 411 is adjusted to remove your statement that your data is NMAR (or alternatively, please prove the statement).

1) Masic I. Predatory Publishing - Experience with OMICS International. Med Arch. 2017;71(5):304–307. doi:10.5455/medarh.2017.71.304-307

2) Wei R, Wang J, Su M, et al. Missing Value Imputation Approach for Mass Spectrometry-based Metabolomics Data. Sci Rep. 2018;8(1):663. Published 2018 Jan 12. doi:10.1038/s41598-017-19120-0

3) Do, K.T., Wahl, S., Raffler, J. et al. Metabolomics (2018) 14: 128. <https://doi.org/10.1007/s11306-018-1420-2>

Reviewer #4 (Remarks to the Author):

Michonneau et al presented a manuscript titled “Metabolomics in human acute Graft-versus-Host Disease”. They applied high throughput metabolomics in two cohorts of genotypically HLA-identical related recipient and donor pairs. The results suggested that microbiota-derived metabolites especially AhR ligands, bile acids and plasmalogens may play significant roles on GvHD. This is a very interesting paper, especially with regard to the two independent cohorts used which clearly added value to findings of the study.

Major comments

My main concerns are related to data analysis methods used with regard to the objectives of the study. It seems that authors mainly applied univariate approaches (t-test and fold changes) followed by “manually” going through the biological roles of significant compounds. Here are my comments each main steps below

1. “First step: Selection of metabolites, imputation of undetectable amounts” There seems to be significant efforts in missing value imputations and evaluating frequencies of detection. These are common issues in metabolomics and although there is no perfect solution here, there are widely used approaches to deal with this. For instance, filtering metabolites with 50% missing and replace remaining with ½ of the detection limits are the default procedures for the popular MetaboAnalyst tool. In authors’ approach “adding some Gaussian noise” which may marginally address bias (which needs to be validated if this is meaningful at all).

2. "Second step: Comparison of average amounts for each chemical"

Here authors refer to using bilateral Students' T tests. Based on the statements L383-385, it seems authors can use bilateral Students' T tests for confounder adjustment. Please clarify the "bilateral" here as this is my first time hear this term (not shown up in my Google search). In L440, there is mentioning of using STAT package (but no citation). I searched online and find this R package "STAT: Interactive Document for Working with Basic Statistical Analysis". This is clearly a general framework to help with learning R, rather than specific methods or parameters used in analysis.

For selections of significant metabolites, please confirm that different p value cutoffs used for the two cohorts are because of the Bonferroni corrections. If so, it is much straightforward to simply state that cutoffs are based on Bonferroni corrected p values 0.05

3. "Third step: Global comparison and identification of similarly behaving chemicals" Here the authors squarely focused on using a "network" based approach to work on ratios rather than common abundance values. This is a new method recently published for qRT-PCR and RNA-Seq (Curis et al. Jan. 2019) but relatively unknown to the metabolomics field. I recommend authors at least first try other widely used multivariate methods like PCA/PLSDA which also show global comparisons and similarly behaving chemicals. It will be more convincing to show that the "network"-based method is indeed better comparing to these simple widely used approaches.

4. "Fourth step: metabolites predicting subjects' group" This title clearly suggesting to me this is biomarker analysis. it could be of great interest to create biomarker models for prediction and report its performance using cross validation as well as test on independent cohort (cohort2). But further reading I found out that authors only tried to identify predictive metabolites, rather than performing the predictions and evaluating the performance per se. The large number of metabolites and small number of samples are common issues in omics biomarker analysis in which logistic regressions may not work well. However, more modern multivariate and machine learning methods (PLS-DA, SVM, randomforests) are able to deal with this issue reasonably well. Authors should at least try these

5. "Pathway analysis"

I only see p-values for individual metabolites, but don't see p-values associated with pathways. It is important to show that a pathway is significantly involved (i.e. not by random chance). In Figure 1c, there are clear pathway information for all metabolites measure. It is intuitive to apply pathway enrichment analysis (such as over-representation analysis) on these compounds to obtain p values.

The MetaboAnalyst web server offers almost all the methods mentioned above. The authors mentioned using the MetaboAnalystR package in their Reporting Summary (Software and Code), but it is not clear how it was used as it was not cited nor mentioned within the manuscript text.

Other minor comments:

- L79: “...Liquid Chromatography-Tandem Mass Spectrometry (UPLC/MS)” => UPLC-MS/MS
- Figure legends – they are too long, should be more concise
- Online Methods. It seems that a lot of text (i.e. L274- L354) were on the well-established and proprietary technologies by Metabolon. I think this level of details is only necessary if the method used were developed specifically for this study.
- It is now generally recommended that metabolomics data should be deposited to either EBI MetaboLights (<https://www.ebi.ac.uk/metabolights/>) or Metabolomics Workbench (<https://www.metabolomicsworkbench.org>)

Response to the Reviewers' comments:

Please note: In blue & italic are modifications that have been introduced in the text and that are highlighted in yellow in the revised manuscript

Reviewer #1:

Appreciate the good faith effort in answering the critiques adequately.

We thank again reviewer 1 for his insightful comments. Of note, we added in the discussion a paper recently published in Nature Biotechnology demonstrating, using the LASSO tool, how metabolome can predict gut microbiome alpha-diversity. The following sentences were added to the discussion with a new reference (Wilmanski et al, Nature Biotechnology, 2019):

“Recently, it was demonstrated that plasma metabolome can predict gut microbiome α -diversity⁵⁶. In this study, 40 metabolites were identified as being associated with human disease and microbiome, many of them being also identified after transplantation, in our study.”

Reviewer #3:

The amendments that the authors have made to the first version strengthen the paper. I have only a few minor comments to make:

We thank reviewer 3 for these new insightful comments. Please find our responses and corrections for these minor points.

1- Line 170: Thank you for adding the point about dietary changes which add an additional perspective. I was in fact referring to the potential for dietary differences between the Ra and the Rs groups leading to the observed differences – this may have a causal factor, but may equally be a confounding factor in data interpretation.

This is a pertinent comment and that’s why we were very careful to artificial feeding (parenteral nutrition) that could bias our results, especially in patients with GvHD (Ra group). However, as patients of Ra group were sampled before any GvHD treatment, they still have a normal oral feeding, quite similar to those of the Rs group (no GvHD). Only one patient in each group had enteral or parenteral feeding (See supplementary table 1).

2- Line 266: what was the ratio or amount of methanol used? What was the concentration of recovery standards used?

Metabolites are extracted in 80% methanol; this has now been added to methods. The standards are not used for quantitation so we considered that their concentrations were not a relevant parameter.

3- Line 282: what were the concentrations of internal standards used? Were they used only for retention time indexing or also to normalize data?

The internal standards are not used for quantitation, they were used strictly for instrument monitoring, so we considered that their concentrations were not a relevant parameter for the methods section.

4- Line 298: The paper you mention you have added to the manuscript does not seem to be either in the reference list or referred to in the methods section. The identically titled Bridgewater 2014 paper is listed in the references but not referenced in the text. I could find no specific reference to a previously published method. Please add the reference if wished (see note below) and include the percentage of each solvent and the solvent gradients used.

NB. The reference you gave has also been wrongly cited as Evans AM, Bridgewater BR, Liu Q, Mitchell MW, Robinson RJ, et al. (2014) High Resolution Mass Spectrometry Improves Data Quantity and Quality as Compared to Unit Mass Resolution Mass Spectrometry in High-Throughput Profiling Metabolomics. *Metabolomics* 4:132. doi:10.4172/2153-0769.1000132.

This gives the impression it has been published in the highly regarded “Metabolomics” journal published by the noted academic publishing group Springer. The only paper I could find that matches the reference above was published in “Metabolomics: Open Access”, an Omics International journal of unknown reputation. The Bridgewater 2014 paper has also been published by this group. I believe Omics International is regarded by many as a predatory publisher (e.g., see Masic et al (1)). While this does not exclude that they may be excellent papers scientifically, there may be a question mark about the amount and quality of any peer review that they underwent. If you choose to use this reference, please update the journal name to reflect the correct journal.

We agree that it is necessary to describe methods as precisely as possible. To avoid any confusion about methods, we decided to remove this reference (previously referred as number 58) from our manuscript and we now completed methods section. We hope that description of methods will be satisfactory you and that you will find all required information (added modifications in blue italic).

“The following information were added to the section methods, in “mass spectrometry” paragraph: One aliquot was analyzed using acidic positive ion conditions (*LC pos*), chromatographically optimized for more hydrophilic compounds. In this method, the extract was gradient eluted from a C18 column (Waters UPLC BEH C18-2.1x100 mm, 1.7 μm) using water and methanol, containing 0.05 % perfluoropentanoic acid (PFPA) and 0.1% formic acid (FA) at *pH=2.5*. *Elution was performed at 0.35mL/min in a linear gradient from 5% to 80% of methanol containing 0.1% FA and 0.05% PFPA over 3.35 minutes*. A second aliquot was also analyzed using acidic positive ion conditions: however it was chromatographically optimized for more hydrophobic compounds. In this method, the extract was gradient eluted from the same afore mentioned C18 column using *methanol 50%, acetonitrile 50%, water, 0.05 % PFPA and 0.01 % FA at pH=2.5* and was operated at an overall higher organic content. *Elution was performed at 0.60mL/min in a linear gradient from 40% to 99.5% over 1 minute, hold 2.4 minutes at 99.5% of methanol 50%, acetonitrile 50%, 0.05 % PFPA and 0.01 % FA*. A third aliquot was analyzed using basic negative ion optimized conditions with a separate dedicated C18 column (*LC neg*). The basic extracts were gradient *eluted from the column using methanol 95% and water 5%, with 6.5mM ammonium bicarbonate at pH 8*. *Elution was performed at 0.35mL/min with a linear gradient from 0.5% to 70% of methanol 95%, water 5% with 6.5mM ammonium bicarbonate over 4 minutes, followed by a rapid gradient to 99% in 0.5 minutes*. The sample injection volume was 5 μL and a 2x needle loop overfill was used. Separations utilized separate acid and base-dedicated 2.1 mm x 100 mm Waters BEH C18 1.7 μm columns held at 40°C. *The fourth aliquot was analyzed via negative ionization following elution from a HILIC column (LC HILIC) (Waters UPLC BEH Amide 2.1x150 mm, 1.7 μm, held at 40°C) using a gradient consisting of water (15%), methanol (5%) and acetonitrile (80%) with 10 mM ammonium formate, pH 10.16*. *Elution flow rate was 0.5mL/min with a linear gradient from 5% to 50% in 3.5 minutes, followed by a linear gradient from 50% to 95% in 2 minutes, of water (50%), acetonitrile (50%) with 10mM ammonium formate, pH 10.6*. The MS analysis alternated between MS and data-dependent MSⁿ scans using dynamic exclusion. The scan range varied slightly between methods but covered 70-1000 *m/z*.”

5- Supp Table 1. I would recommend defining NA as meaning not available, as opposed to not applicable to avoid confusion.

Thanks for this comment; we now defined *NA as not available in the footnote to the table*.

6- Line 318: no reference or details for this cocktail of QC standards is given. Please also define the abbreviation MTRX.

The list of internal standards utilized for the LC Neg, LC HILIC, and LC Pos methods is shown in the table below and has been added to the “quality assurance and quality control” paragraph:

Method	Internal Standards
LC Neg	D7-glucose, d3-methionine, d3-leucine, d8-phenylalanine, d5-tryptophan, bromophenylalanine, d15-octanoic acid, d19-decanoic acid, d27-tetradecanoic acid, d35-octadecanoic acid, d2-eicosanoic acid
LC HILIC	D35-octadecanoic acid, d5-indole acetic acid, bromophenylalanine, d5-tryptophan, d4-tyrosine, d3-serine, d3-aspartic acid, d7-ornithine, d4-lysine,
LC Pos	d7-glucose, d3-methionine, d3-leucine, d8-phenylalanine, d5-tryptophan, bromophenylalanine, d4-tyrosine, d5-indole acetic acid, d5-hippuric acid, amitriptyline, d9-progesterone, d4-dioctylphthalate

MTRX (abbreviation for “matrix samples”) refers to samples generated from a pool of human plasma extensively characterized by Metabolon, Inc. For plasma-based studies, Metabolon utilizes MTRX samples as technical replicate controls. Overall process variability is determined by calculating the median RSD for all endogenous metabolites (i.e., non-instrument standards) present in these samples.

This definition has now been clarified in the same paragraph:

“Quality assurance and quality control (QA/QC)

Several types of controls were analyzed in concert with the experimental samples: a pooled matrix sample generated by taking a small volume of each experimental sample (or alternatively, use of a pool of well-characterized human plasma, *named MTRX for “sample matrix”*) served as a technical replicate throughout the data set; extracted water samples served as process blanks; and a cocktail of QC *standards listed below*, that were carefully chosen not to interfere with the measurement of endogenous compounds were spiked into every analyzed sample, allowed instrument performance monitoring and aided chromatographic alignment.

Methods	Internal Standards
LC Neg	D7-glucose, d3-methionine, d3-leucine, d8-phenylalanine, d5-tryptophan, bromophenylalanine, d15-octanoic acid, d19-decanoic acid, d27-tetradecanoic acid, d35-octadecanoic acid, d2-eicosanoic acid
LC HILIC	D35-octadecanoic acid, d5-indole acetic acid, bromophenylalanine, d5-tryptophan, d4-tyrosine, d3-serine, d3-aspartic acid, d7-ornithine, d4-lysine,
LC Pos	d7-glucose, d3-methionine, d3-leucine, d8-phenylalanine, d5-tryptophan, bromophenylalanine, d4-tyrosine, d5-indole acetic acid, d5-hippuric acid, amitriptyline, d9-progesterone, d4-dioctylphthalate

Instrument variability was determined by calculating the median relative standard deviation (RSD) for the internal standards that were added to each sample prior to injection into the mass spectrometers (median RSD = 3-4%). Instruments are calibrated at least weekly in the utilized polarity using Thermo and mass accuracy is monitored at the batch level for the internal standards. A batch fails QC if any of the internal standards are more than 5ppm away from the theoretical mass.

Overall process variability was determined by calculating the median RSD for all endogenous metabolites (i.e., non-instrument standards) present in 100% of the MTRX samples, which are

technical replicates created from a large pool of extensively characterized human plasma. The median RSD for the MTRX samples was equal to 9-10%. Five MTRX samples and three process blank samples were processed per every batch of 30 samples. Experimental samples were randomized across the platform run with QC samples spaced evenly among the injections.”

7- Line 334 onwards. The identifying list of MSMS is a key point to being able to reproduce the experimental conditions. Without them being listed, no reader has the opportunity of being able to review any possible problems with identification, especially of key metabolites. Ideally the full list of identifying features should be published in supplementary, at least for the metabolites listed as being of key importance. Alternatively, you could reference another peer-reviewed article where this has already been done (the references provided do not give enough details). I understand the commercial problems this request may cause for Metabolon. This must be an editorial decision as to whether the current methods included here is an acceptable compromise.

Due to the proprietary nature of the Metabolon chemical spectral library, we are unable to comply with the request to provide full information on identifying MSMS features as Supplementary Data. However, we added the full list of metabolites including mass and retention time index as supplementary data, that we hope, will help readers to determine how metabolites were identified and to reproduce data. The Metabolomics Standards Initiative outlines the minimum standards for a Level 1 identification (the highest confidence level of identification) as: “*A minimum of two independent and orthogonal data relative to an authentic compound analyzed under identical experimental conditions are proposed as necessary to validate non-novel metabolite identifications.*” (Sumner et al). Thus, by providing the mass and RI, even without MS/MS fragmentation, we hope to meet the highest standard needed to validate the metabolite identification. This information is now provided in supplementary table 13.

Sumner LW, Amberg A, Barrett D, Beale MH, Beger R, Daykin CA, Fan TW, Fiehn O, Goodacre R, Griffin JL, Hankemeier T, Hardy N, Harnly J, Higashi R, Kopka J, Lane AN, Lindon JC, Marriott P, Nicholls AW, Reily MD, Thaden JJ, Viant MR. (2007) Proposed minimum reporting standards for chemical analysis Chemical Analysis Working Group (CAWG) Metabolomics Standards Initiative (MSI). *Metabolomics* 3(3): 211-221. doi: 10.1007/s11306-007-0082-2

8- Line 366: I presume that the data handling for D v Rs is identical as to that for D v Ra. Please update the text to reflect.

Yes, this is identical for both comparisons, we corrected the manuscript.

9- Line 411: Lastly, there are no easy answers to the perennial problem of missing values. I agree entirely with you that with 50% missing values, leaving out data would be equally problematic; we tend to have a much higher initial filter when we adopt this method. Thank you for bringing the interesting paper by Keizer et al to my attention. The data concerns pharmacokinetic data, rather than metabolomics, and the highest number of samples they had below the LOQ was 40% which they had determined was a „very high“ number of missingness. I notice that they conclude that likelihood-based imputation performs better than LLOQ/2. This is a similar conclusion to other papers in the field e.g. Wei et al 2019(2) and Do et al 2018(3).

We really thank the reviewer for these references and the discussion about the recurrent problem of missing data in metabolomics studies.

- First, we note that after filtering metabolites with more than 50 % missing values either globally or in at least one of the groups, only a few remaining metabolites presented a relatively high percentage of missing value [for instance, there are four metabolites with more than 40 % missing values for the Rs vs Ra comparison – caprylate (8.0), glycodeoxycholate, glycohyocholate and N-acetylglucosaminy-asparagine]. Hence, this is likely that the results of Keizer et al. on high missingness would only affect the results marginally.

- In addition, most of the metabolites finally selected are metabolites that do not present missing values; the four previous metabolites, for instance, were neither detected by the univariate approach, nor by the network approach - but they were selected by the PLS-based methods in one of the cohorts.

All of these points suggest that the handling of these missing data is likely not to markedly modify the overall conclusions of the paper. This is, by the way, in agreement with the findings of Do et al. on p. 128, fig. 7A: all confidence intervals do overlap, so in practice all methods lead roughly to the same global conclusion. Otherwise, the more extreme results concern two variants of complex imputation methods, that is, MTIS-R and MICE-norm (single case of non-overlap, but it could typically also be a potential multiple-comparisons issue).

Nevertheless, to further check the impact of the imputation method, we have performed several sensitivity analyses, using complete case analysis and simple imputation methods with different fractions of the minimum value as reported in Do et al. (25 %, 50 % = min/2, 75 %, 90 %, 99 %, and 100 % = min). Indeed, since our relatively small dataset, a complex imputation method like fitting a truncated (log-) Gaussian distribution or multiple imputations as well as random forest appeared inappropriate – for instance, regarding metabolites with “as low” as 20 % missing values, only 12 patients had a GVH, and this makes about 10 values to fit such a complex model. Whatever the imputation method, the resulting list of selected metabolites has changed only marginally, and the results discussed in the paper have been reported robust to such a sensitivity analysis.

This point has now been added to the manuscript to discuss bias and limit of the imputation methods.

Results of the different methods are not included in the paper but are available upon request for reviewers or readers if necessary. As an example, reviewers will find an example of the univariate analysis comparing recipient with or without GvHD in cohort 1 and 2 using for different methods: missing value imputation with LLOQ/2 + gaussian noise, LLOQ/2 without noise, imputation with median and no imputation (see radar plots below and the enclosed excel files entitled “imputation methods for reviewers.xls”).

We also added the very interesting reference suggested by reviewer 3 from Do et al., metabolomics, 2018, as we considered that it really gives a major reference to go further in the discussion, to readers interested by this subject.

Do et al discuss the issue of MNAR values as being the most likely for their data which, in the context of pharmacokinetics, makes sense. It is also seen in many metabolomics datasets for logical reasons. I have also dealt with data where ion suppression effects meant that the majority of missing data was missing at random (MAR). With your specific data, the correlation plots of MV versus intensity that you printed in your response were difficult to fully interpret as the y axes scale was a little stretched. However, there appeared to be no correlation between intensity and missingness by your own statistical analysis as answer to point 4 in the previous review (correlation coefficients of -0.1 and -0.05). Allowing for the possibility that some metabolites may be highly intense in some individuals and almost non-existent in others, the results would still point to missing values having a high proportion of MAR values – hence suggesting that other approaches such as random forest or k-nearest neighbor imputation would be preferable to half minimum value. Perhaps the compromise situation is to draw the readers' attention to the potential bias of the approach used and highlight the percentage of missing values, the average detected intensity (excluding missing values) and the limit of detection for each class for metabolites found to be of major significance. I would recommend that the new paragraph you added at 411 is adjusted to remove your statement that your data is NMAR (or alternatively, please prove the statement).

1) Masic I. Predatory Publishing - Experience with OMICS International. *Med Arch.* 2017;71(5):304–307. doi:10.5455/medarh.2017.71.304-307

2) Wei R, Wang J, Su M, et al. Missing Value Imputation Approach for Mass Spectrometry-based Metabolomics Data. *Sci Rep.* 2018;8(1):663. Published 2018 Jan 12. doi:10.1038/s41598-017-19120-0

3) Do, K.T., Wahl, S., Raffler, J. et al. *Metabolomics* (2018) 14:128. <https://doi.org/10.1007/s11306-018-1420-2>

Distinction between MNAR and MAR process is quite difficult, and there is no statistical method that can select between those two assumptions: in other words, these are actually untestable assumptions. Therefore, only external knowledge of the mechanisms leading to missing data can help to choose between these two underlying settings.

First, in our opinion, our data are basically MNAR: it is the fact that the metabolite concentration is low that makes it undetected. Even if the group membership may help to predict the missingness for some metabolites, it is the metabolite's value itself that triggers or not the missingness. Nevertheless, such extreme cases have been handled in our analysis by the comparison of missingness proportion between the groups.

Moreover, to consider the possibility of a MAR mechanism, making our imputation by min/2 possibly severely biased, we have performed sensitivity analyses based on simple imputation using the mean or the median. Nevertheless, as reported above (see our answer to the previous point), we do think that complex imputation methods, such as random forests or nearest neighbours, cannot easily apply to such a small dataset. The very similar results of these imputations, close to those of the primary analyses, confirm that, after the filtering of highly missing metabolites, our analysis procedure was not markedly influenced by the handling of missing data.

As the distinction between MNAR and MAR is not proved, we deleted the sentence *“Such missingness process actually depends on missing values, suggesting missing not at random (MNAR) mechanism of missing data.”*, and modified the following sentence to avoid any confusion for readers:

“Based on the experimental protocol, missing data can be considered mainly to rely on the limit of quantification (LOQ) of the analytical method: any value below this limit could not be measured and reported (BLQ data). To consider the information of such BLQ data – known as “left-censoring”-, the most usual method is to impute half the LoQ to missing data.”

To help readers in their interpretation of our results, we also added: the average detected intensity, without missing value imputation, the percentage of missing value, and the minimal detected value to table 1.

Reviewer #4:

Michonneau et al presented a manuscript titled “Metabolomics in human acute Graft-versus-Host Disease”. They applied high throughput metabolomics in two cohorts of genotypically HLA-identical related recipient and donor pairs. The results suggested that microbiota-derived metabolites especially AhR ligands, bile acids and plasmalogens may play significant roles on GvHD. This is a very interesting paper, especially with regard to the two independent cohorts used which clearly added value to findings of the study.

Major comments

My main concerns are related to data analysis methods used with regard to the objectives of the study. It seems that authors mainly applied univariate approaches (t-test and fold changes) followed by “manually” going through the biological roles of significant compounds. Here are my comments each main step below

1. “First step: Selection of metabolites, imputation of undetectable amounts”

There seems to be significant efforts in missing value imputations and evaluating frequencies of detection. These are common issues in metabolomics and although there is no perfect solution here, there are widely used approaches to deal with this. For instance, filtering metabolites with 50% missing and replace remaining with ½ of the detection limits are the default procedures for the popular MetaboAnalyst tool. In authors’ approach “adding some Gaussian noise” which may marginally address bias (which needs to be validated if this is meaningful at all).

We agree that the most popular approach is a simple imputation of censored observations with ½ of the detection limits. However, adding a small random noise to the imputed value should avoid the tie issue. Nevertheless, to confirm that adding that Gaussian noise had not any strong influence on the results, we have rerun the same analyses without adding this noise. Results were qualitatively the same (that is, the same metabolites were selected, whatever the different methods, despite slightly different *p*-values), suggesting that such a noise had no marked effect on the results and conclusions of the paper.

2. “Second step: Comparison of average amounts for each chemical”

Here authors refer to using bilateral Students’ T tests. Based on the statements L383-385, it seems authors can use bilateral Students’ T tests for confounder adjustment. Please clarify the “bilateral” here as this is my first time hear this term (not shown up in my Google search). In L440, there is mentioning of using STAT package (but no citation). I searched online and find this R package “STAT: Interactive Document for Working with Basic Statistical Analysis”. This is clearly a general framework to help with learning R, rather than specific methods or parameters used in analysis.

We apologize for this confusion. We should have used “two-tailed Student T-test”, instead of “bilateral” which is the French term for that. This has been corrected in the revised manuscript.

The package used is “stats”, not STAT; “stats” is a basic package of R (<https://www.R-project.org/>) that allows to perform Student’s T test (t.test function), exact Fisher test for contingency tables (fisher.test function), and hierarchical clustering (hclust function) among many other standard statistical methods that were used in the analyses presented in this paper. We apologize for the typo. Since this package is distributed along with R and given potential confusion for readers, we have removed mention to it in the revised manuscript.

For selections of significant metabolites, please confirm that different p value cutoffs used for the two cohorts are because of the Bonferroni corrections. If so, it is much straightforward to simply state that cutoffs are based on Bonferroni corrected p values 0.05

Yes, different cut-off values came from Bonferroni correction with a different number of tests. While it indeed corresponds to corrected p -values < 0.05 , p -values reported in the text and table are raw p -values. For this reason, we think that for the reader, cut-offs defined in the raw p -values scale are more readily usable. Moreover, this lets the reader free to choose the multiplicity correction method (especially adapt the number of tests to its own convenience, or implement a Holm, or any other correction procedure), and avoids the replacement of several p -values by 1 when corrected. This has been now stated more clearly in the text. We added the following sentence to methods section, "second step" paragraph:

"Consequently, $p < 8.77 \times 10^{-5}$ (that is, $p^ < 0.05$ where p^* is p after Bonferroni correction) was used as the threshold to detect significantly differentially present metabolites"*

3. "Third step: Global comparison and identification of similarly behaving chemicals"

Here the authors squarely focused on using a "network" based approach to work on ratios rather than common abundance values. This is a new method recently published for qRT-PCR and RNA-Seq (Curis et al. Jan. 2019) but relatively unknown to the metabolomics field. I recommend that authors at least first try other widely used multivariate methods like PCA/PLSDA which also show global comparisons and similarly behaving chemicals. It will be more convincing to show that the "network"-based method is indeed better comparing to these simple widely used approaches.

We wish to thank the Reviewer for this suggestion. We have performed a PCA (using the R package FactoMineR, function PCA), with the first axes allowing to separate Rs and Ra patients, suggesting a set of important metabolites strongly contributing to these axes. We have also performed a PLS-DA (using the R package mixOmics, function plsda), that clearly separates Rs and Ra patients, and tried to select metabolites using a sparse version of PLS-DA (package mixOmics in R, function splsda). Results of these analyses have been now reported in the paper. However, we would kindly argue that these analyses are not completely satisfactory, for the following reasons.

First, PCA and PLS-DA assume independent samples. While this may be considered true for the Rs vs Ra comparison, this is an arguable assumption for the comparisons of donors versus recipients.

Secondly, PCA and PLS-DA in their usual implementation apply to unconstrained data; however, metabolomics results are, by nature, compositional (due to the normalization steps, typically). Special variants of PCA and PLS-DA exist for compositional data, but are hard to use when there is a mix of compositional (metabolite amounts) and non-compositional (age, sex, BMI...) data, making the inclusion of covariates difficult. However, compositional variants of PLS-DA and ACP here gave very similar results than their standard variants when no covariate was added, so this aspect probably has little impact here.

Last, and this is the main drawback in our case, PLS-DA is known to *always* find a separation between groups when the number of variates is much higher than the number of samples (see for instance Wehrens, R. 2011. *Chemometrics with R: Multivariate Data Analysis in the Natural Sciences and Life Sciences*; Springer). Here, with a sample size around 60 patients and around 600 metabolites, it is typically a case where it is difficult to trust PLS-DA results. Cross-validation could alleviate this issue: however with only 12 Ra patients in one cohort, it is difficult to perform a meaningful cross-validation study.

The network-based method, we already used, does not suffer from these drawbacks: the cut-off is selected by previous simulation (not using the data) to control the error rate of spurious findings, the all-pairwise tests method is meant for handling compositional data and, last, since the test used for each edge removal can be anything, including covariates or pairing is straightforward.

This has been discussed in the revised manuscript. A new figure (figure 6) representing PCA, PLS-DA and ORA results was added to main results, together with the list of newly identified metabolites (table 2) and the following paragraph:

“To confirm the implication of these metabolites at acute GvHD onset, PCA was used to confirm that recipient with or without GvHD could be discriminated in multivariate analysis (figure 6a and b). Metabolites that mostly contribute to acute GvHD profile were then identified with sparse PLS-DA (figure 6c and d, table 2) and used to build an over-representation analysis of the main pathways that contribute to GvHD (figure 6e and f). This approach confirmed that most of metabolites identified in univariate analysis were also selected by sPLS-DA and belong to the previously identified metabolic pathways, especially plasmalogens, arginine and tryptophan metabolism.”

The corresponding methods were added to the “data analysis” paragraph in the online methods:

“At the end of this step, a principal component analysis (PCA) and a sparse Partial Least Square Discriminant Analysis (sPLS-DA) were performed, to check that the different groups were indeed separated, allowing identifying metabolites whose level differ between the groups (R package FactoMineR, function PCA and R package mixOmics, function plsda and splsda). Metabolites that were identified in sPLS-DA were then used to build an over-representation analysis (ORA). Enrichment (E) was calculated by considering the number of metabolites identified with sPLS-DA in each pathways (k), the total number of metabolites identified in sPLSD-DA (n), the number of metabolites in each pathway (m) and the total number of metabolites used for analysis (M) as follow: $E=(k/m)/((n-k)/(N-m))$. For each pathway, p value was determined by calculation of the hypergeometric distribution. “

4. “Fourth step: metabolites predicting subjects’ group”

This title clearly suggesting to me this is biomarker analysis. It could be of great interest to create biomarker models for prediction and report its performance using cross validation as well as test on independent cohort (cohort2). But further reading I found out that authors only tried to identify predictive metabolites, rather than performing the predictions and evaluating the performance per se. The large number of metabolites and small number of samples are common issues in omics biomarker analysis in which logistic regressions may not work well. However, more modern multivariate and machine learning methods (PLS-DA, SVM, randomforests) are able to deal with this issue reasonably well. Authors should at least try these

We are sorry that this step’s subtitle led to confusion. We have changed the title to *“Fourth step: metabolite selection using group membership modelling”*.

However, we would like to stress that the aim of the study is not to predict if patients develop or not GVHD. To perform such a predictive study, samples should have been taken at the time of graft infusion (before GVH disease) and not at disease onset. Other (proteomic) GVHD biomarkers have already been developed by others in large cohorts. Instead, the aim of the study was to identify metabolites at acute GVHD onset with the aim to better understand the underlying mechanisms of the disease. Hence, the approach is to use a variable selection process to identify these metabolites, but not to predict if a patient has (or will have) GVHD.

As a consequence, PLS-DA, random forests and SVM that are not meant to perform variable selection do not answer our question. As mentioned above, sparse PLS-DA was performed per Reviewer 4 request. On the other hand, logistic regression with LASSO variable selection is perfectly suited to do the intended task, and was used for this purpose.

Finally, accuracy of group membership prediction does not appear a pertinent marker of the quality of the selected metabolites set. Instead, the overlap of the selected metabolites sets in both cohorts, or of their metabolic pathways, is (in our view) a more pertinent marker of the different groups with regard to the pathophysiology of GVHD.

5. “Pathway analysis”

I only see p-values for individual metabolites, but don't see p-values associated with pathways. It is important to show that a pathway is significantly involved (i.e. not by random chance). In Figure 1c, there are clear pathway informations for all metabolites measure. It is intuitive to apply pathway enrichment analysis (such as over-representation analysis) on these compounds to obtain p values.

The MetaboAnalyst web server offers almost all the methods mentioned above. The authors mentioned using the MetaboAnalystR package in their Reporting Summary (Software and Code), but it is not clear how it was used as it was not cited nor mentioned within the manuscript text.

We thank the reviewer for this comment and we agree that an over representation analysis is an interesting method to identify pathway that are involved in GvHD. As mentioned in point 3, **we now added an ORA to the new figure 6 (e and f) and in the results section**. Using this approach, we confirmed that plasmalogens, arginine and tryptophan pathways were among the most enriched and significant pathways involved at acute GvHD onset.

Other minor comments:

- **L79: “Liquid Chromatography-Tandem Mass Spectrometry (UPLC/MS)” => UPLC-MS/MS**

This has been corrected in the manuscript.

- **Figure legends – they are too long, should be more concise**

We agree that figure legends were long as we tried to describe methods in details. We now tried to reduce legends length by removing informations that were available in the method section but we kept enough information to allow each figure to be interpreted independently.

- **Online Methods. It seems that a lot of text (i.e. L274- L354) was on the well-established and proprietary technologies by Metabolon. I think this level of details is only necessary if the method used were developed specifically for this study.**

We thank you for this comment; however, it was the editor's recommendation to fully describe technical aspects in the methods section.

- **It is now generally recommended that metabolomics data should be deposited to either EBI MetaboLights (<https://www.ebi.ac.uk/metabolights/>) or Metabolomics Workbench (<https://www.metabolomicsworkbench.org>)**

This has been done and all raw data are now available on MetaboLights and registered as MTBLS204 / www.ebi.ac.uk/metabolights/mtbls204 (cohort 1) and MTBLS205 / <https://www.ebi.ac.uk/metabolights/mtbls205> (cohort 2).

For journal and reviewers only, the following links are available during the reviewing process:

MTBLS204: <https://www.ebi.ac.uk/metabolights/reviewersd6eafd8b422e879868420021516fc3e0>

MTBLS205: <https://www.ebi.ac.uk/metabolights/reviewers375fdb3d2fae83a24670785c67c41eb5>

REVIEWERS' COMMENTS:

Reviewer #3 (Remarks to the Author):

The authors have done their best to balance the need for scientific openness and maintain commercial competitiveness and have addressed the vast majority of the points I have raised. I am happy to recommend the paper for publication.

One final point: Reviewer 4 requested the use of PLS-DA which the authors have duly complied with. PLS-DA and related methods are supervised techniques where overfitting is common. I would strongly advise the authors to always include information about their model validation here.

Reviewer #4 (Remarks to the Author):

The authors have sufficiently addressed my comments in their revised manuscript. I only have one minor comment: please cite those software packages used in your data analysis (mixOmics, factoMineR, etc) - the formal acknowledgement in your citation is important for the long-term development and support of these tools

REVIEWERS' COMMENTS:

Reviewer #3 (Remarks to the Author):

The authors have done their best to balance the need for scientific openness and maintain commercial competitiveness and have addressed the vast majority of the points I have raised. I am happy to recommend the paper for publication.

One final point: Reviewer 4 requested the use of PLS-DA which the authors have duly complied with. PLS-DA and related methods are supervised techniques where overfitting is common. I would strongly advise the authors to always include information about their model validation here.

We thank reviewers for their insightful comments and for their interest in our manuscript.

We definitely agree with Reviewer 3 that PLS-DA is prone to overfitting. In fact, when the number of predictors is much higher than the number of samples (as is the case in our study), PLS-DA will always find a hyperplane separating the two groups. This is one of the reasons why we first did not perform this analysis, and we added it to comply with Reviewer 4 request. Another reason is that PLS-DA does not by itself allow to conveniently identify which predictors are informative, that is in our application which metabolites are changed between the two groups.

Unfortunately, our small sample size does not allow to perform a meaningful crossvalidation study. However, the fact that PLS-DA leads to similar results on two independent cohorts is by itself an external validation, suggesting that the separation may not come only from overfitting.

The fact that metabolites identified by PLS-DA (more exactly its variante, sparse PLS-DA) show a good overlap with metabolites identified by other methods, less prone to overfitting, is also an indirect confirmation of the PLS-DA results.

However, and we tried to state it clearly in the paper, PLS-DA is not our "reference" method, but is just given for illustrative purposes. The conclusions given by the paper are not derived from the PLS-DA results, we just mention that PLS-DA results are in agreement with the conclusion drawn from other methods.

For this reason, we decided to accept the Reviewer 4 suggestion to present PLS-DA results despite its validation is at best difficult in this setting.

The following sentence was added to the manuscript to clarify this point: *"Due to the small sample size and the very low (sample number)/(predictor number) ratio, PLS-DA is prone to overfitting and cross-validation cannot be used here. Hence, PLS-DA results should be taken as descriptive."*

Reviewer #4 (Remarks to the Author):

The authors have sufficiently addressed my comments in their revised manuscript. I only have one minor comment: please cite those software packages used in your data analysis (mixOmics, factoMineR, etc) - the formal acknowledgement in your citation is important for the long-term development and support of these tools

We perfectly agree with this comment from reviewer 4, and R packages used in this study are now, not only cited in Methods, but original article references were also added to bibliography:

63. Lê, S., Josse, J. & Husson, F. FactoMineR: An R Package for Multivariate Analysis. *J. Stat. Softw.* **25**, 1–18 (2008).
64. Rohart, F., Gautier, B., Singh, A. & Lê Cao, K.-A. mixOmics: An R package for 'omics feature selection and multiple data integration. *PLoS Comput. Biol.* **13**, e1005752 (2017).
65. Curis, E. *et al.* Determination of sets of covarying gene expression using graph analysis on pairwise expression ratios. *Bioinformatics* **35**, 258–265 (2019).
66. Tibshirani, R. Regression Shrinkage and Selection via the Lasso. *J. R. Stat. Soc. Ser. B Methodol.* **58**, 267–288 (1996).